# BAdam: A Memory Efficient Full Parameter Optimization Method for Large Language Models

**Qijun Luo**[1,2]    **Hengxu Yu**[1]    **Xiao Li**[1*]

[1]The Chinese University of Hong Kong, Shenzhen
[2]Shenzhen Research Institute of Big Data
{qijunluo,hengxuyu}@link.cuhk.edu.cn, lixiao@cuhk.edu.cn

## Abstract

This work presents BAdam, an optimization method that leverages the block coordinate descent (BCD) framework with Adam's update rule. BAdam offers a memory efficient approach to the full parameter finetuning of large language models. We conduct a theoretical convergence analysis for BAdam in the deterministic case. Experimentally, we apply BAdam to finetune the Llama 3-8B and Llama 3-70B models using a single RTX3090-24GB GPU and $4\times$A100-80GB GPUs, respectively. The results confirm BAdam's efficiency in terms of memory usage, running time, and optimization capability. Furthermore, the downstream performance evaluation based on MT-bench and math benchmarks shows that BAdam outperforms existing memory efficient baselines such as LoRA. It also demonstrates that BAdam can achieve comparable or even superior performance compared to Adam. Finally, the ablation study using SGD's update rule illustrates the suitability of BCD for finetuning LLMs. Our code can be easily integrated into any PyTorch-based codebase and is available at https://github.com/Ledzy/BAdam.

## 1 Introduction

Large language models (LLMs) such as GPT-4 [1] and Llama 3 [15] have shown its strong ability in language understanding, generation, reasoning, translation, etc [5, 73, 72, 60]. Due to its strong applicability, LLMs have been regarded as a feasible approach towards artificial general intelligence [6]. Finetuning or adaptation has become an important step in applying pretrained LLMs to follow human instructions or perform specific downstream tasks [44, 63].

**Backgrounds.** When GPU memory (RAM) is not a major limitation, full parameter finetuning methods—such as applying Adam to the entire set of parameters of LLMs—often offer the most flexibility for parameter search. However, executing such a full parameter training method typically requires a significant amount of GPU memory. For instance, to finetune an LLM with $M$ billion parameters, Adam [23] necessitates roughly $18M$ GB of GPU memory for successful training, and this estimate does not even account for the storage of activations used in the backpropagation (BP) process; see Section 2.2.1 for a detailed analysis. This requirement poses challenges for computational resources as models scale up, given the fact that GPU memory is often limited in practical settings.

Parameter efficient finetuning (PEFT) methods such as low-rank adaptation (LoRA) [22], Adapter [21], prompt- and prefix-tuning [29, 26], among others, play a critical role in finetuning large language models under memory resource constraints. The principal idea of PEFT is to represent the parameter updates in a much lower-dimensional subspace and, consequently, the memory consumption is significantly reduced. Despite the success of PEFT methods, finetuning within a substantially lower-dimensional subspace may potentially limit downstream performance; see, e.g., [62].

---

[*]Corresponding Author

38th Conference on Neural Information Processing Systems (NeurIPS 2024).

| Method | Memory | Full parameter training | Momentum and second moment | Update precision | Gradient accumulation |
|--------|--------|-------------------------|----------------------------|------------------|------------------------|
| Adam [23] | $18M$ | ✓ | ✓ | Float32 | ✓ |
| LOMO [37] | $2M + \frac{2M}{D}$ | ✓ | ✗ | Float16 | ✗ |
| LoRA [22] | $2M + \frac{36rM}{m}$ | ✗ | ✓ | Float32 | ✓ |
| BAdam | $2M + \frac{16M}{D}$ | ✓ | ✓ | Float32 | ✓ |

Table 1: Algorithm feature summary. Here, $M$ represents that the model to be trained has $M$ billion number of parameters, $r$ is the LoRA rank, $m$ is the weight matrix dimension (here, we consider square weight matrices for simplicity), $D$ is the number of transformer layers in LOMO or the number of partitioned blocks in BAdam. BAdam performs full parameter mixed precision update, while only requires memory that is comparable to LOMO and LoRA.

The observations outlined above motivate us to explore a memory efficient full parameter optimization method without imposing low-rank constraint on the parameter update.

**Main results.** In this work, we have the following main contributions:

(C.1) We propose a *block coordinate descent (BCD)-type* optimization method with Adam's update rule, termed BAdam; see Section 2.1 for the detailed description. This method partitions the entire set of model parameters into $D$ blocks, updating one block at a time using Adam's efficient update steps. BAdam offers a memory efficient solution to the full parameter finetuning of LLMs. For example, by partitioning a model with $M$ billion parameters into $D$ equal-sized blocks, BAdam requires only about $2M + \frac{16M}{D}$ GB of GPU memory for successful mixed precision training; see Section 2.2.1 for detailed analysis. This leads to a significant reduction in memory demands compared to full parameter finetuning using Adam. Theoretically, we provide a convergence analysis for BAdam in the deterministic case, demonstrating that leveraging the BCD framework and Adam's update rule yields a convergent scheme; see Theorem 2.1.

(C.2) We apply BAdam to finetune the Llama 3-8B and Llama 3-70B models using *a single RTX3090-24GB GPU* and $4 \times A100\text{-}80GB$ *GPUs*, respectively. Specifically, we present in Section 3.1 BAdam's efficiency in both memory consumption and running time. In Section 3.2, we empirically verify BAdam's optimization capability via its fast convergence and the high rankness of its learned perturbations. We further evaluate the downstream performance of different methods using MT-bench and several math benchmarks; see Section 3.3. The results illustrate that BAdam generally outperforms existing memory efficient baselines such as LoRA. Importantly, BAdam achieves comparable average performance with Adam on math benchmarks and even surpasses Adam in instruction-following tasks evaluated by MT-bench score. Moreover, we conduct ablation study using SGD's update rule (BSGD) in Section 3.4. The results show that BCD variants maintain optimization capability compared to their full counterparts and even exhibit better downstream performance. It also demonstrates that BSGD can achieve similar downstream performance to BAdam, illustrating the effectiveness and suitability of BCD for finetuning LLMs.

We compare BAdam with several representative methods in Table 1. In summary, we believe that BAdam may serve as a viable alternative optimization method to state-of-the-art methods such as LoRA in scenarios with limited computing memory.

## 2 The BAdam Method

Block coordinate descent (BCD) method has a long history in optimization society, which can be traced back to the very origins of the discipline; see, e.g., [43, 36, 4, 55, 41, 58]. At each iteration, BCD maintains the majority of the optimization parameters at their up-to-date iteration values, while it approximately optimizes the objective function over the remaining parameters, resulting in a much lower dimensional problem.

BCD is known to be efficient for huge-scale problems where the number of optimization parameters is extensive [41], particularly when it significantly exceeds the number of data points / component

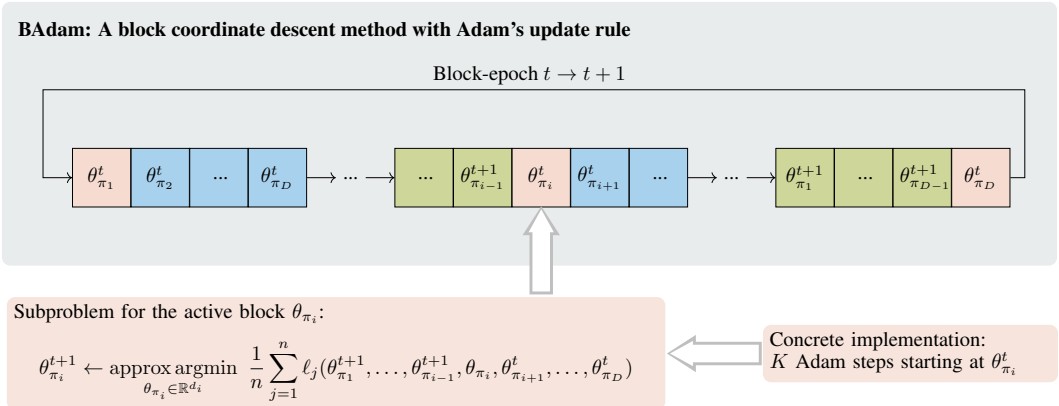

Figure 1: Illustration of the proposed BAdam, which is based on the block coordinate descent framework. Colors represent the states of the partitioned blocks in one block-epoch, including the active block, non-updated inactive blocks, and updated inactive blocks.

functions. Based on this main feature, we reveal an interesting link between BCD and the finetuning of LLMs. Namely, the finetuning process boils down to an optimization problem that needs to handle a huge number of trainable model parameters, while the number of training data points are relatively small. This setting matches exactly the advantage of the BCD scheme, providing the possibility to release the requirement on large GPU memory. We refer to Sections 3.2 and 3.4 for empirical verification on the power and suitability of BCD for finetuning LLMs.

## 2.1 Algorithm Description and Convergence Result

In this subsection, we propose BAdam, a block coordinate descent method embedded with Adam's update rule. The method is displayed in Algorithm 1 and illustrated in Figure 1. Formally, let us consider an abstract formulation for finetuning LLMs $\min_\theta \mathcal{L}(\theta) = \frac{1}{n} \sum_{j=1}^n \ell_j(\theta)$. Here, $\theta \in \mathbb{R}^d$ represents the concatenation of the vectorized parameters of the model, $n$ is the number of training data points, and $\ell_j$ is the negative log-likelihood loss function for language modeling on the $j$-th training data point.

**Block partition and block coordinate descent framework.** At the $t$-th block-epoch, BAdam first generates an ordered block partition $\pi = \{\pi_1, \ldots, \pi_i, \ldots, \pi_D\}$, which splits the whole model parameters $\theta \in \mathbb{R}^d$ into $D$ blocks, i.e., $\theta = \{\theta_{\pi_1}, \ldots, \theta_{\pi_i}, \ldots, \theta_{\pi_D}\}$ with $\theta_{\pi_i} \in \mathbb{R}^{d_i}$ and $\sum_{j=1}^D d_j = d$. The partition $\pi$ can be very flexible and is a unified representation. Given a large language model, one natural block partition is by transformer modules. Apart from this partition, one can also choose a small part of parameters from each transformer module and regard these parameters as one block $\theta_{\pi_i}$. Note that $\pi$ may be either a deterministic or a random partition, as long as the aggregation of all the blocks $\{\theta_{\pi_i}\}$ forms the whole set of parameters $\theta$. For instance, if we partition by consecutive transformer modules, we may list the blocks in $\pi$ in ascending order (e.g., from the input to the output module), descending order (e.g., from the output to the input module), or random reshuffling order.

We now present the optimization framework of BAdam. Our core idea is to adopt the main spirit of BCD. Namely, *we approximately optimize over only one active block $\theta_{\pi_i}$ at one time, given that the other inactive blocks are fixed at their up-to-date values*. Mathematically, at the $t$-th block-epoch, updating the current active block $\theta_{\pi_i}$ amounts to solving the following subproblem:

$$\theta_{\pi_i}^{t+1} \leftarrow \operatorname*{approx\,argmin}_{\theta_{\pi_i} \in \mathbb{R}^{d_i}} \frac{1}{n} \sum_{j=1}^n \ell_j(\theta_{\pi_1}^{t+1}, \ldots, \theta_{\pi_{i-1}}^{t+1}, \theta_{\pi_i}, \theta_{\pi_{i+1}}^t, \ldots, \theta_{\pi_D}^t). \tag{1}$$

When this approximate minimization becomes exact, this scheme is also known as Gauss-Seidel method or alternating minimization. Even with exact minimization, some literature still refers to it as BCD. One can see that subproblem (1) fixes the inactive blocks at their most recent values, and hence

---

**Algorithm 1:** BAdam: A block coordinate descent method with Adam's update rule.

---

**1** **input:** $\beta_1$, $\beta_2$, $\varepsilon$, $K$, and learning rate $\alpha$.

**2** **initialization:** block-epoch index $t \leftarrow 0$ and model parameters $\theta^0$.

**3** **while** *stopping criterion not meet* **do**

**4**     generate a block partition $\pi = \{\pi_1, \cdots, \pi_D\}$ ;

**5**     **repeat** *for one block-epoch* $i \leftarrow 1, \ldots, D$                       // BCD loop

**6**        $k \leftarrow 0$;    $m_{\pi_i}^{t,0} \leftarrow 0$;    $v_{\pi_i}^{t,0} \leftarrow 0$;    $\theta_{\pi_i}^{t,0} \leftarrow \theta_{\pi_i}^t$ ;       // Block initialization

**7**        **repeat** *for $K$ Adam steps to update the active block* $\theta_{\pi_i}$

**8**           $k \leftarrow k + 1$;

             // compute the block stochastic gradient

**9**           $g_{\pi_i}^{t,k} \leftarrow$ stochastic approx. of $\frac{\partial}{\partial \theta_{\pi_i}} \mathcal{L}(\theta_{\pi_1}^{t+1}, \ldots, \theta_{\pi_{i-1}}^{t+1}, \theta_{\pi_i}^{t,k-1}, \theta_{\pi_{i+1}}^t, \ldots, \theta_{\pi_D}^t)$;

**10**           $m_{\pi_i}^{t,k} \leftarrow \beta_1 m_{\pi_i}^{t,k-1} + (1-\beta_1)g_{\pi_i}^{t,k}$,    $v_{\pi_i}^{t,k} \leftarrow \beta_2 v_{\pi_i}^{t,k-1} + (1-\beta_2)(g_{\pi_i}^{t,k})^2$ ;

**11**           $\hat{m}_{\pi_i}^{t,k} \leftarrow m_{\pi_i}^{t,k}/(1-\beta_1^k)$,           $\hat{v}_{\pi_i}^{t,k} \leftarrow v_{\pi_i}^{t,k}/(1-\beta_2^k)$ ;

**12**           $\theta_{\pi_i}^{t,k} \leftarrow \theta_{\pi_i}^{t,k-1} - \alpha \hat{m}_{\pi_i}^{t,k}/\left(\sqrt{\hat{v}_{\pi_i}^{t,k}} + \varepsilon\right)$ ;         // Adam update

**13**        **end**

**14**        $\theta_{\pi_i}^{t+1} \leftarrow \theta_{\pi_i}^{t,K}$;

**15**        $g_{\pi_i}, m_{\pi_i}, v_{\pi_i} \leftarrow$ **None** ;          // clear memory for grad and optim states

**16**     **end**

**17**     $t \leftarrow t + 1$;

**18** **end**

**19** **return** learned model parameters $\theta^t$.

---

it is a much lower dimensional optimization problem compared to $\min_\theta \frac{1}{n} \sum_{j=1}^n \ell_j(\theta)$, providing the possibility to implement the method in situations with limited GPU memory. Solving subproblem (1) sequentially for $i = 1, \ldots, D$ moves the block-epoch from $t$ to $t + 1$.

**Update using Adam steps.** Similar to most of the concrete BCD methods, we propose to implement the approximate minimization subproblem (1) using several gradient-based steps starting at $\theta_{\pi_i}^t$. Abstractly, BAdam executes the update

$$\theta_{\pi_i}^{t+1} \leftarrow \mathcal{A}(\theta_{\pi_1}^{t+1}, \ldots, \theta_{\pi_{i-1}}^{t+1}, \theta_{\pi_i}^t, \theta_{\pi_{i+1}}^t, \ldots, \theta_{\pi_D}^t). \tag{2}$$

We choose the algorithmic procedure $\mathcal{A}$ in (2) to be $K$ Adam steps [23] starting at $\theta_{\pi_i}^t$, in order to efficiently decrease the objective function. To specify the concrete Adam steps, we first note that the gradient of the objective function can be correspondingly decomposed as

$$\nabla \mathcal{L}(\theta) = \begin{bmatrix} \frac{\partial \mathcal{L}}{\partial \theta_{\pi_1}} & \cdots & \frac{\partial \mathcal{L}}{\partial \theta_{\pi_D}} \end{bmatrix}^\top = \begin{bmatrix} \frac{\partial}{\partial \theta_{\pi_1}} \frac{1}{n} \sum_{i=1}^n \ell_i(\theta) & \cdots & \frac{\partial}{\partial \theta_{\pi_D}} \frac{1}{n} \sum_{i=1}^n \ell_i(\theta) \end{bmatrix}^\top. \tag{3}$$

We call $\frac{\partial \mathcal{L}}{\partial \theta_{\pi_i}}$ in (3) the block gradient of the objective function $\mathcal{L}$ over block $\theta_{\pi_i}$. According to the main spirit of stochastic optimization methods, we select a batch of data points to compute a block stochastic gradient $g_{\pi_i}$ using the up-to-date iterates for approximating the block gradient, as outlined in Line 9 of Algorithm 1. With $g_{\pi_i}$, we construct the momentum and second moment for the active block $\theta_{\pi_i}$ as shown in Line 10 – Line 11. Finally, we implement Adam update in Line 12. One may also invoke decoupled weight decay [34] into Line 12. In summary, Line 6 – Line 15 concretely implement the BCD update (1).

It is important to note that BAdam differs from existing BCD with momentum approaches [41], which often maintain dense momentum vectors. BAdam is specifically designed for memory efficiency, and hence it clears the optimizer states in Line 15. Additionally, we do not offload the optimizer states, as they will no longer correspond to the updated block parameters in the next block-epoch. Thus, clearing the states in Line 15 and starting with zero initial states for every new active block in Line 6 are crucial for ensuring convergence, stability, and memory efficiency of our method.

The number $K$ in Line 7 of Algorithm 1 is the only additional hyperparameter introduced by BAdam, compared to Adam. We provide a detailed discussion on selecting $K$ in Appendix B.2.

**Convergence result.** We provide a convergence analysis for BAdam in the deterministic case, aiming to establish that combining the block coordinate descent framework with Adam's update rule results

in a convergent scheme. We consider the extension to the stochastic case as future work. Indeed, combining the analysis for Adam with stochastic gradients, as in [65, 27, 57], with our analysis for the block coordinate descent framework could be a feasible direction for such an extension. The informal theorem is presented below, while the formal theorem and proofs are put in Appendix D.

**Theorem 2.1** (informal). BAdam *using deterministic gradients is a descent method, under certain commonly utilized conditions for analyzing block coordinate descent method and Adam. That is, after one block-epoch of updates for the whole model, we have*

$$\mathcal{L}(\theta^{t+1}) - \mathcal{L}(\theta^t) \leq -\mathcal{O}(\alpha K)\|\nabla\mathcal{L}(\theta^t)\|^2. \tag{4}$$

*Consequently,* BAdam *finds a $\delta$-approximate stationary point within $\mathcal{O}(\delta^{-2})$ iterations.*

We conclude this section by noting that BAdam is essentially a block coordinate descent method, in which the BCD framework achieves low memory consumption. Apart from the chosen Adam's update rule, it is possible to propose other efficient optimization procedures for concretely implementing (1); see Section 3.4 for an ablation study where we also employ SGD's update rule.

## 2.2 Analysis of Memory Consumption and BP Time

### 2.2.1 Memory Consumption Analysis

We analyze the memory consumption of BAdam, caused by storing the model parameters, gradient, and optimizer states. Let us consider a large language model with $M$ billion parameters. We will use GB as the unit of GPU memory in the sequel.

We first analyze the memory cost of Adam with mixed precision training. One needs to store the FP16 model parameters for the BP process, which costs $2M$ memory. For a more precise update, the optimizer also maintains a master copy of a FP32 model, which costs $4M$ memory. Then, it comes to store the gradient (converted to FP32), momentum, and second moment in FP32 precision, costing $4M + 4M + 4M = 12M$ memory. In total, Adam needs roughly $18M$ memory.

In terms of BAdam, it needs to store the up-to-date model parameters (see Figure 1) in FP16 precision, which costs $2M$ memory. Importantly, since BAdam only updates the active block at one time, we can store the model parameters, gradient, momentum, and second moment *only for the active block* $\theta_{\pi_i}$ in FP32 precision, where the FP32 model parameters and gradient of the active block can be obtained by transforming their FP16 versions to the FP32 versions. Let us consider the simple case where the partitioned $D$ blocks are equal-sized. Then, BAdam only needs in total

$$\mathbf{2M} + \frac{\mathbf{16M}}{\mathbf{D}} \text{ memory.} \tag{5}$$

Note that the above analyses do not account for the memory required to store activations, as this is associated with the BP process rather than the optimization method itself. Furthermore, gradient checkpointing [11] can be employed to reduce the memory requirement needed for storing activations. We display the actual memory consumption for finetuning the Llama 3-8B model in Section 3.1.

### 2.2.2 BP Time Analysis for Consecutive Module-based Block Partition

We consider the specific case where the partitioned $D$ blocks $\{\theta_{\pi_i}\}$ are $D$ consecutive transformer modules of LLMs. Thanks to the property of backpropagation, BAdam can reduce the computation time of BP compared to Adam and LoRA under the same amount of data utilization.

Let us consider one block-epoch of BAdam, meaning that it has trained with $K \cdot D$ data batches, where $K$ is defined in Algorithm 1. We consider that each data point has the same sequence length and each transformer module has the same amount of parameters, in order to ease the analysis. Recall that a BP process consists of a forward pass and a backward pass. For the forward pass, BAdam has almost the same computational load as that of Adam, while LoRA requires more forward computation due to its extra low-rank adapters. Hence, it remains to consider the number of unit-backward-pass after utilizing $KD$ data batches, where the unit-backward-pass is defined as a backward pass of a single data batch through a single transformer module. Importantly, BAdam only updates the active block, and hence the number of unit-backward-pass largely depends on the depth of the active block. For instance, if the input module or output module is the current active block, we need $D$

unit-backward-pass or only 1 unit-backward-pass, respectively. Thus, after one block-epoch (i.e., utilizing $KD$ data batches), BAdam requires

$$K(1 + \cdots + D) = \frac{KD(D + 1)}{2} \quad \text{unit-backward-pass.} \tag{6}$$

However, Adam and LoRA need to backward for all the $D$ transformer modules, thus requiring $KD^2$ unit-backward-pass after utilizing $KD$ data batches.

Apart from saving the number of unit-backward-pass, some of the unit-backward-pass of BAdam may even take less computational time compared to that of Adam. Let us take the backward pass of the input module as an example. BAdam does not require explicit stochastic gradient computation of the model parameters of the intermediate modules $\partial z_l / \partial \theta_l$, where $\{z_l\}$ are the activations of the intermediate modules and $\{\theta_l\}$ are the trainable model parameters of these modules. However, Adam needs to compute these quantity explicitly. We refer to Table 4 for an experiment illustration.

In summary, BAdam with consecutive module-based block partition saves computational load of the BP process compared to Adam and LoRA, after training with the same amount of data. We demonstrate this through experiments detailed in Section 3.1. If the module-based block partition is not consecutive, for instance, when one block consists of modules (such as matrices) from different transformer layers, we still expect that BAdam can reduce BP time to some extent, though not as significantly as indicated by (6).

## 3 Experiment Results

In this section, we evaluate the proposed BAdam on finetuning LLMs. Selected baselines include LOMO (essentially SGD) [37], LoRA [22], Galore [66], and Adam [23]. All BAdam experiments for training the Llama 2-7B and Llama 3-8B models are conducted on a single RTX3090-24GB GPU, whereas BAdam experiments for the Llama 3-70B model use $4 \times$ A100-80GB GPUs. Experiments for the baseline methods are conducted using either a single RTX3090 or multiple A100 GPUs, depending on their memory requirements. Our implementation is based on Llama-Factory [69]. Detailed experiment setup can be found in Appendix B.1.

### 3.1 Memory Consumption and Wall-clock Running Time

In this subsection, we present the empirically measured memory consumption and wall-clock running time of BAdam and baseline methods. All the measurements in this subsection are based on finetuning the Llama 3-8B model on Alpaca-GPT4 dataset [46] using a single RTX3090-24GB GPU.

**Memory consumption.** We report the actual memory consumption of BAdam and the baseline approaches in Table 2 for finetuning the Llama 3-8B model, in which the memory consumption of Adam is estimated rather than tested. This result indicates that all of LOMO, LoRA (with a reasonable rank), and BAdam can finetune the Llama 3-8B model using a single RTX3090. It can be observed that all the methods have nearly the same memory cost for storing the model parameters, while LoRA requires slightly more memory due to its low-rank adapters. Furthermore, LOMO, LoRA, and BAdam significantly reduce memory consumption regarding the storage of the gradient and optimizer states compared to Adam. Moreover, it is easy to see that the total memory consumption (the last column of Table 2) is higher than the sum of the listed quantities. The additional memory costs arise from storing activations and training data, pre-allocated memory caches by PyTorch, and other buffers for intermediate computing results. Indeed, our tests show that BAdam can successfully finetune the Llama 3-8B model with input sequences of length 1024 using a batch size of 2, or input sequences of length 2048 using a batch size of 1, with a single RTX3090-24GB GPU.

**Wall-clock running time comparison.** We conduct experiments on finetuning the Llama 3-8B model for 3 epochs with each method and report the averaged wall-clock time per epoch; see Table 3. The forward time for three approaches are rather close. The slightly higher time cost for LOMO and LoRA attributes to additional operations for registering activations and the calculation of the low-rank adapters, respectively. Regarding backward time, BAdam reduces the time cost by nearly half compared to LoRA and LOMO, supporting the analysis in Section 2.2.2. It is important to note that the backward time for all methods includes the re-forward time due to gradient checkpointing, which diminishes the running time advantage of BAdam.

| Method | Parameter | Gradient | Optimizer states | Memory consumption |
|--------|-----------|----------|------------------|--------------------|
| Adam | 16.1GB | 32.1GB | 96.6GB | 144.8GB+ |
| LOMO | 16.1GB | 0.5GB | — | 21.5GB |
| LoRA-rank100 | 16.7GB | 1.0GB | 3.1GB | OOM |
| LoRA-rank8 | 16.2GB | 0.1GB | 0.3GB | 22.3GB |
| BAdam | 16.1GB | 0.9GB | 2.6GB | 23.5GB |

Table 2: Actual memory costs of applying mixed precision training to finetune Llama 3-8B with gradient checkpointing using a single RTX3090. Note that LOMO only supports FP16 precision training. The maximum input sequence length is 728 and the batch size is 2.

| Method | Forward | Backward | Update |
|--------|---------|----------|--------|
| LOMO | 0.78 hours | 3.70 hours | — |
| LoRA | 0.83 hours | 3.20 hours | 136 seconds |
| BAdam | 0.71 hours | 1.74 hours | 142 seconds |

| Backward scheme | Backward time |
|-----------------|---------------|
| All modules | 0.64 seconds |
| Input module only | 0.33 seconds |
| Output module only | 0.03 seconds |

Table 3: Time spent per epoch on forward, backward, and update for finetuning Llama 3-8B using a single RTX3090. The single pass batch size is 2. The results are averaged over 3 epochs.

Table 4: Time spent on different backward schemes with batch size 2 for finetuning Llama 3-8B using a single RTX3090. The results are averaged over 100 backward passes.

In Table 4, we conduct tailored experiments to further support our analysis in Section 2.2.2. It can be observed that: 1) backward for "Output module only" is almost time free, as it requires only 1 unit-backward-pass; 2) backward for "All modules" takes significantly more time, as it has to implement $D$ unit-backward-pass; and 3) backward for "Input module only" takes less time than $D$ unit-backward-pass (i.e., backward for "All modules"), since the former scheme does not need to compute the stochastic gradients of the intermediate modules' parameters.

## 3.2 Optimization Capability

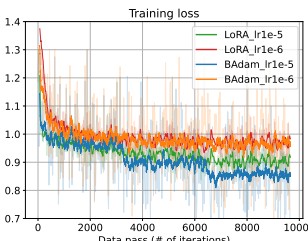 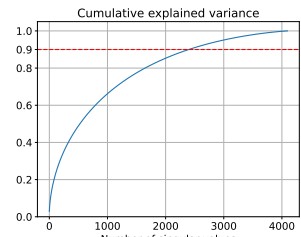 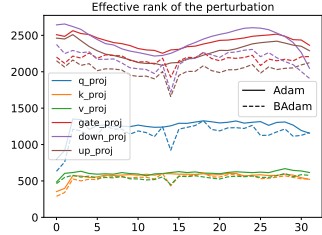

Figure 2: Optimization capability of BAdam for finetuning Llama 3-8B on Alpaca-GPT4 dataset. Left: Online training loss of LoRA and BAdam. Middle: Cumulative explained variance of BAdam's learned perturbation to the 25th layer's up-proj matrix. Right: Effective rank of Adam's and BAdam's learned perturbations.

We verify the optimization capability of BAdam through both the training loss convergence and the effective rank of the learned perturbations. Experiments in this subsection correspond to exactly the same training process of the lower block of Table 5 for finetuning the Llama 3-8B model.

**Loss convergence.** In the left figure of Figure 2, we display the online training loss. From a pure optimization perspective, namely, in terms of driving the training loss lower, BAdam demonstrates better convergence behavior than LoRA when using 1e-5 as the initial learning rate. If the initial learning rate is set to 1e-6, BAdam initially converges slightly faster, but the two methods soon align as the learning rate becomes too small to make substantial progress.

**Effective rank of the learned perturbations.** We empirically measure the learning and optimization capability of BAdam through the effective rank of its learned perturbations, i.e., the difference between the learned weight matrix and the pretrained base weight matrix $\Delta W := W_K - W_0$. The

cumulative explained variance "cvar" and the effective rank of matrix $A \in \mathbb{R}^{m \times n}$ are defined as:

$$\text{cvar}(r) := \frac{\sum_{i=1}^{r} \sigma_i(A)^2}{\sum_{j=1}^{\min\{m,n\}} \sigma_j(A)^2}, \quad \text{effective\_rank}(A) := \min\{r : \text{cvar}(r) \geq 0.9\},$$

where $\sigma_i(A)$ is the $i$-th largest singular value of $A$.

In the middle figure of Figure 2, we display cvar of BAdam's update for the 25th layer's up-proj matrix. This result shows that BAdam's update has a heavy tailed singular values distribution and is far away from a low-rank update. In the right figure of Figure 2, we plot the effective rank of the learned perturbation by BAdam and Adam through all modules of different transformer layers. Notably, BAdam achieves almost the same high rank update as Adam, which partly justifies BAdam's learning and optimization capability.

## 3.3 Downstream Performance Evaluation

In this subsection, we conduct supervised finetuning for the Llama 2-7B [54], Llama 3-8B, and Llama 3-70B models on the Alpaca-GPT4 and MathInstruct [61] datasets. The setting of hyperparameters is deferred to Appendix B.2.

**MT-bench results.** To illustrate the models' downstream performance, we report the MT-bench scores of the instruction-tuned models obtained by different methods for 3 epochs. We utilize two initial learning rates, 1e-5 and 1e-6, with a cosine scheduler for all methods. The results are displayed in Table 5.

| Model: Llama 2-7B (base model MT-bench: 3.93) | | | | | | | | | | |
|---|---|---|---|---|---|---|---|---|---|---|
| **lr** | | | 1e-5 | | | | | 1e-6 | | |
| **Method** | Adam | LOMO | LoRA | Galore | BAdam | Adam | LOMO | LoRA | Galore | BAdam |
| Epoch 1 | 4.41 | 4.01 | 4.77 | 4.70 | 4.79 | 4.62 | 3.99 | 4.59 | 4.12 | 4.71 |
| Epoch 2 | 4.73 | 4.06 | 4.84 | 4.83 | 5.21 | 4.94 | 4.02 | 4.86 | 4.17 | 4.83 |
| Epoch 3 | 5.16 | 4.11 | 4.01 | 4.88 | 4.87 | 5.13 | 4.06 | 4.81 | 4.26 | 4.88 |
| **Average** | 4.76 | 4.06 | 4.54 | 4.80 | **4.96** | **4.90** | 4.02 | 4.75 | 4.18 | 4.81 |

| Model: Llama 3-8B (base model MT-bench: 5.46) | | | | | | | | | | |
|---|---|---|---|---|---|---|---|---|---|---|
| **lr** | | | 1e-5 | | | | | 1e-6 | | |
| **Method** | Adam[a] | LOMO | LoRA | Galore | BAdam | Adam | LOMO | LoRA | Galore | BAdam |
| Epoch 1 | – | 5.49 | 6.17 | 5.78 | 6.07 | 6.15 | 5.40 | 6.41 | 5.66 | 6.65 |
| Epoch 2 | – | 5.62 | 6.36 | 5.80 | 6.19 | 6.26 | 5.85 | 6.19 | 5.77 | 6.64 |
| Epoch 3 | – | 5.41 | 6.28 | 5.89 | 6.64 | 6.29 | 5.83 | 6.20 | 5.70 | 6.67 |
| **Average** | – | 5.51 | 6.27 | 5.82 | **6.30** | 6.23 | 5.69 | 6.27 | 5.71 | **6.65** |

Table 5: MT-bench scores of the instruction-tuned Llama 2-7B and Llama 3-8B on Alpaca-GPT4 dataset by different methods.
[a] Adam with learning rate 1e-5 for finetuning Llama 3-8B overfits the Alpaca-GPT4 dataset and achieves MT-bench scores that are even lower than that of the base model. Hence, we omit this outlier.

Some remarks and observations on Table 5 are in order. 1) Using 1e-5 as the initial learning rate, the average MT-bench score over 3 epochs achieved by BAdam surpasses that of LoRA by a magnitude of **0.42** for instruction-tuning the Llama 2-7B model. Regarding instruction-tuning the Llama 3-8B model using an initial learning rate of 1e-6, the average score returned by BAdam outperforms that of LoRA by a magnitude of **0.38**. 2) In most cases, BAdam can beat LoRA and Galore, albeit sometimes slightly, across the two learning rate settings and when evaluating checkpoints from different epochs for both the Llama 2-7B and Llama 3-8B models. This underscores the promising performance of our proposed method. 3) BAdam is on par with the performance of Adam for the Llama 2-7B model and outperforms Adam for the Llama 3-8B model, partly illustrating the power of the BCD optimization scheme in LLM finetuning. It is worth noting that BAdam is both memory and running time efficient. In terms of memory usage, it requires only a single RTX3090-24GB GPU for finetuning the Llama 3-8B model, while Adam needs multiple A100-80GB GPUs.

| Model: Llama 3-8B | | | | | | | |
|---|---|---|---|---|---|---|---|
| **Method** | GSM8K | Aqua | MMLU-Math | SAT-Math | MATH | NumGLUE | **Average** |
| Base model | 25.9 | 22.8 | 33.7 | 39.5 | 12.8 | 34.5 | 28.2 |
| Adam | **54.5** | 40.5 | 44.3 | 51.4 | **18.4** | 55.4 | 44.1 |
| LOMO | 32.1 | 28.0 | 40.0 | 39.5 | 13.1 | 37.1 | 31.6 |
| LoRA | 47.5 | **44.9** | 45.3 | 50.9 | 14.5 | **56.9** | 43.3 |
| Galore | 33.1 | 37.4 | 41.2 | 42.7 | 15.0 | 36.9 | 34.4 |
| BAdam | 48.1 | 42.5 | **50.5** | **56.8** | 15.7 | 53.0 | **44.4** |
| Model: Llama 3-70B | | | | | | | |
| **Method** | GSM8K | Aqua | MMLU-Math | SAT-Math | MATH | NumGLUE | **Average** |
| Base model | 52.4 | 46.5 | 52.2 | 58.2 | 21.2 | 37.9 | 44.7 |
| LoRA | 73.3 | 59.5 | 58.3 | 64.1 | **34.2** | 64.8 | 59.0 |
| BAdam | **78.8** | **63.4** | **64.2** | **76.4** | 26.2 | **67.3** | **62.7** |

Table 6: Zero-shot math benchmark scores of the finetuned Llama 3-8B and Llama 3-70B on MathInstruct dataset by different methods.

**Math benchmarks.** We also finetune the Llama 3-8B and Llama 3-70B models on the MathInstruct dataset for 3 epochs, and evaluate the trained model using math benchmarks across different domains. The results are shown in Table 6. In terms of average score, BAdam outperforms all the memory efficient baselines, and even slightly surpasses the benchmark score of Adam while requiring significantly less memory consumption compared to Adam. In particular, for the experiments on finetuning Llama 3-8B, BAdam outperforms LoRA in 4 out of 6 tasks, and surpasses LOMO and Galore in all the tasks by a large margin. For finetuning Llama 3-70B, BAdam beats LoRA in 5 out of 6 tasks.

### 3.4 Ablation Study: SGD's Update Rule

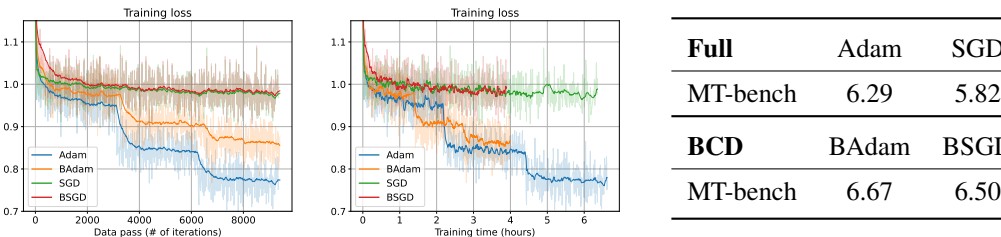

| **Full** | Adam | SGD |
|---|---|---|
| MT-bench | 6.29 | 5.82 |

| **BCD** | BAdam | BSGD |
|---|---|---|
| MT-bench | 6.67 | 6.50 |

Figure 3: Ablation study for BCD variants and their full counterparts for finetuning Llama 3-8B on Alpaca-GPT4 dataset. Left and middle: Convergence behavior. Right: MT-bench scores.

In this subsection, we conduct ablation study to consider SGD's update rule in our BCD framework, leading to BCD with SGD (BSGD). Then, we compare the performance of BAdam, BSGD and their full counterparts, i.e., Adam and SGD, to illustrate the power of BCD in LLMs finetuing.

**Optimization.** In the left and middle figures of Figure 3, we display the training loss of BAdam, BSGD, and their full counterparts. It can be observed that BCD variants converge slightly slower but soon exhibit similar convergence behavior in terms of running time compared to their full counterparts. It is worth mentioning that, unlike the full counterparts, BCD only updates one block of parameters per data batch, demonstrating the strong optimization ability of BCD for LLMs finetuning.

**Downstream performance.** In the right table of Figure 3, we test the MT-bench scores of the four methods. It is quite interesting to see that BSGD significantly outperforms SGD (almost as good as BAdam), even though they have almost the same optimization convergence behavior. We suspect that the superiority of the BCD variants over their full counterparts possibly stems from the fact that BCD uses each data batch to update only one block of parameters, thereby better preserving the general knowledge of the pretrained model during finetuning. These improved downstream performance of BCD compared to their full counterparts further illustrate its suitability for LLM finetuning.

### 3.5 Additional Experiment Results

We provide more experiment results in Appendix C. Here are the summarized results: 1) In Appendix C.1, we conduct an ablation study on the ordering scheme of the partition $\pi$ in BAdam, considering random reshuffling, ascending, and descending orders. In terms of convergence behavior, these three choices are competitive. We also provide an ablation study on the hyperparameter $K$ in BAdam, with $K$ being chosen from $\{10, 50, 100, 200\}$. The results indicate that these four choices of $K$ perform similarly in terms of convergence behavior. However, we observe that the convergence speed of choosing different $K$ can vary across different models. We refer to Appendix B.2 for a detailed discussion on the selection of $K$. 2) In Appendix C.2, we examine BAdam's capability in classification tasks by training RoBERTa-large on SuperGLUE benchmarks. The results show that BAdam can achieve similar average scores as Adam. 3) In Appendix C.3, we conduct a preliminary continue pretraining (CPT) experiment. We apply BAdam to train the Llama 3.1-8B-Instruct model on the StarCoder-Python dataset [28] for about 1 epoch. The result shows that BAdam can effectively decrease the CPT loss, making it a strong candidate for CPT tasks when GPU memory is limited. 4) We display the memory consumption and running time costs for finetuning the Llama 2-7B model in Appendix C.4, which match the results for finetuning the Llama 3-8B model presented in Section 3.1.

In summary, our experiment results in Section 3 demonstrate that BAdam has the potential to serve as a competitive optimization method for finetuning LLMs when the GPU memory is limited, compared to state-of-the-art memory efficient methods such as LoRA.

## 4   Brief Literature Review

We briefly review several memory efficient finetuning methods in this section. A more comprehensive literature review is presented in Appendix A due to limited space.

One major branch for memory efficient finetuning of LLMs is parameter-efficient finetuning (PEFT), which freezes the pre-trained weight and only trains the additional injected parameters. LoRA [22], adapter [21], and prefix-tuning [29] belong to this class and have been verified to be effective in finetuning LLMs. Another line of works focus on memory efficient full parameter finetuning. MeZO [38] performs zero-th order SGD update without calculating the stochastic gradient, thereby only requires the memory of performing inference. LOMO [37] efficiently performs on-the-fly SGD update during the backward pass without storing the stochastic gradient. However, LOMO's implementation design prevents it from using gradient accumulation technique. Galore [66] reduces memory consumption by projecting the gradient into low-rank space. It requires constantly performing SVD to obtain the low-rank projector.

## 5   Conclusion and Discussions on Limitations

In this work, we have proposed the BAdam optimization method, which is built upon the block coordinate descent framework with Adam's update rule. We finetune the Llama 3-8B and Llama 3-70B models on the Alpaca-GPT4 and MathInstruct datasets by BAdam with a single RTX3090-24GB GPU and $4\times$A100-80GB GPUs, respectively. The results illustrated the efficacy of BAdam in terms of GPU memory consumption and running time. Empirically, BAdam exhibits better convergence behavior compared to LoRA and learns high rank update. Further downstream performance assessments have demonstrated BAdam's superior performance in instruction finetuning and math finetuning, in comparison to LOMO, LoRA, and Galore. Additionally, BAdam has on par or even better downstream performance compared to Adam. In summary, we believe that BAdam may serve as a viable alternative for finetuning LLMs with limited memory resources.

**Limitations.** Our focus has been on applying BAdam for supervised finetuning. Extending its application to preference optimization represents another opportunity to demonstrate BAdam's capabilities. Moreover, our CPT experiment using BAdam is only preliminary. Exploring extensively BAdam's performance in the CPT setting is an interesting direction. We leave these directions for future improvements.

**Broader impacts.** Our proposed method significantly lowers the barrier to full parameter finetuning of large models for a broader range of researchers. This is a technical algorithmic contribution that does not yield explicit negative societal impacts. However, it carries a risk of misuse.

## Acknowledgments and Disclosure of Funding

The authors thank the reviewers for their insightful comments, which have helped greatly to improve the quality and presentation of the manuscript.

Xiao Li is supported in part by the National Natural Science Foundation of China (NSFC) under grant 12201534, and in part by the Shenzhen Science and Technology Program under grants RCYX20221008093033010 and RCYX20210609103229031.

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

# Contents

# A  More Related Works

We present a review of the relevant literature below. Given the extensive and rapidly growing body of work in this field, it is important to note that the references we include here are not exhaustive.

**Block coordinate descent method.** The block coordinate descent (BCD) method is a well-established algorithmic scheme in the field of optimization [43, 36, 4, 55, 41, 58], which is especially efficient for problems with an exceptionally large number of trainable parameters. Some advanced analyses on its convergence in both convex and nonconvex cases can be found in, e.g., [50, 35, 42, 67, 7, 8] and the references therein. The works [52, 16, 51, 10, 39] also theoretically discuss the effects of different choices of the block update order in BCD and ADMM-BCD variants, which is related to the ordering scheme in the partition $\pi$ of BAdam. BCD has been widely and practically applied in the machine learning area as well. For instance, a specific instance of BCD, known as layer-wise training, has been utilized for neural network training [20, 3, 2, 71, 45]. The sequential minimal optimization technique integrated into LIBSVM [9] is also a BCD-type method.

**Parameter efficient finetuning (PEFT).** An effective strategy for finetuning LLMs is to train a small number of (possibly extra) model parameters, while keeping the majority of the pretrained parameters frozen. Numerous approaches have been proposed and studied along this line of research. For instance, adapter tuning only finetunes the inserted small modules between layers called adapters [21]. Prompt-tuning / prefix-tuning [26, 29] attaches additional trainable prefix tokens to the input and/or hidden layers, while remaining the base model unchanged. Another prevalent method is the low-rank adaptation (LoRA) [22], which models the increment to the base model as a product of two significantly lower dimensional trainable low-rank matrices. Subsequent research on LoRA has aimed at extending its rank constraints [30, 59], further reducing the number of trainable parameters [24, 25], decreasing memory usage through quantization [14], etc. Presently, LoRA-type methods are commonly employed for finetuning LLMs with limited memory resources. Interested readers are referred to [17] for a unified framework and comprehensive comparison of these PEFT methods.

**Memory efficient full parameter finetuning.** To conduct full parameter finetuning of LLMs with limited memory, the work [37] proposes LOMO, which efficiently leverages the BP process to update parameters on the fly in the process of computing stochastic gradients. Consequently, LOMO helps to execute SGD for full parameter finetuning without physically storing the stochastic gradients, significantly reducing memory consumption. However, it is worth emphasizing that SGD generally converges more slowly and is often considered suboptimal compared to Adam. MeZO [38] is to approximate SGD by using only the forward pass. The idea of MeZO derives from zeroth-order optimization, which utilizes function value difference to approximate the stochastic gradients of the trainable model parameters. Galore [66] uses gradient low-rank projection, which largely reduces memory consumption for full parameter finetuning compared to Adam. Adam-mini [64] proposes to apply block-wise adaptive learning rate, which reduces the memory for storing the full second moment. Another popular approach for finetuning with limited memory is to perform CPU offloads to reduce the memory consumption caused by training data and optimizers; see, e.g., [49, 48, 33].

# B Detailed Experiment Setup and Hyperparameters

## B.1 Experiment Setup

In this subsection, we introduce the setup including the dataset, evaluation, and training details. We present the hyperparameters in Appendix B.2.

**Task setup.** Our experiments mainly consist of instruction tuining, math finetuning, and natural language understanding.

1. **Instruction tuning.** We finetune the Llama 2-7B and Llama 3-8B models on Alpaca-GPT4 dataset [46] for 3 epochs. This dataset consists of 52k instruction-following data generated by GPT-4, using prompts from the Alpaca dataset [53]. The finetuned model is then evaluated using the MT-bench [68] with the "gpt-4" API to test its downstream performance.

2. **Math finetuning.** We finetune the Llama 3-8B and Llama 3-70B models on MathInstruct dataset [61] for 3 epochs, which contains 260K samples from 13 math related datasets. The model is then evaluated on 4 in-domain benchmarks [12, 19, 31, 40] and 2 out-of-domain benchmarks [70, 18] using zero-shot prompt. Our evaluation implementation is based on the released code[2] of [61].

3. **SuperGLUE.** we finetune the RoBERTa-large model [32] with 355 million parameters on the SuperGLUE benchmark [56], and evaluate the performance of the finetuned model using the test dataset. Since the label of the original test dataset is not revealed, we randomly split the "dev" dataset into validation and test dataset; see Table 7. We focus on 6 tasks of the SuperGLUE benchmark, including BoolQ, COPA, WSC, RTE, MultiRC, and WiC. Since the classification modules of RoBERTa-large are randomly initialized, we set these classification modules to be trainable for all methods.

| **Task** | Train | Validation | Test |
|---|---|---|---|
| BoolQ | 9427 | 1270 | 2000 |
| COPA | 400 | 30 | 70 |
| MultiRC | 5100 | 453 | 500 |
| RTE | 2500 | 128 | 150 |
| WiC | 6000 | 238 | 400 |
| WSC | 554 | 44 | 60 |

Table 7: Data split for the SuperGLUE experiment. The original "dev" dataset is randomly splitted into validation and test datasets.

**Setup for different finetuning methods.** For BAdam, we use consecutive module-based block partition represented by transformer layers, resulting in the number of blocks $D = 32$ for the Llama 2-7B and Llama 3-8B models, $D = 80$ for the Llama 3-70B model, and $D = 26$ for the RoBERTa-large model. The ordering strategy in the partition $\pi$ of BAdam is random reshuffling. For Galore, LoRA, and BAdam, we train all the transformer layers while freezing the language modeling head and the embedding layers. For Adam and LOMO, we set all modules in transformer layers to be trainable. We adopt the setup in Galore's paper and apply pure BF16 and 8-bits Adam for all Galore's experiments. We apply pure BF16 precision training for LOMO, as it does not support mixed precision training. Since LOMO does not support gradient accumulation, its batch size is smaller than the other approaches (consequently, it runs more steps) to ensure aligned memory consumption; see Appendix B.2 for more detail.

**Additional implementation details.**

- All the experiments in Section 3.1 are conducted using a single RTX3090-24GB. We use 4×A100-80GB GPUs to finetune the Llama 3-70B model using LoRA and BAdam.

---

[2]https://github.com/TIGER-AI-Lab/MAmmoTH

- For all the experiments of finetuning Llama models, we apply the gradient checkpointing technique [11] to reduce the memory cost of storing activations for all methods. In particular, we checkpoint the input of each transformer layer and re-forward to calculate the layer's activations during the backward phase.

- The experiments on finetuning Llama models are implemented using Llama-Factory [69]. The SuperGLUE experiments are based on the implementation of jiant [47].

## B.2 Hyperparameters

We summarize the choices of hyperparameters for SuperGLUE, instruction tuning, and math finetuning in Table 8, Table 9, and Table 10, respectively. Some discussions follow:

- BAdam introduces only one additional hyperparameter compared to Adam, namely, $K$ in Algorithm 1. One natural choice is $K = \frac{n}{BD}$, where $n$ is the number of training data points, $B$ is the effective batch size, and $D$ is the number of blocks in BAdam. We round $\frac{n}{BD}$ to the nearest integer if it is a fractional. Such a setting ensures that after one block-epoch, all the $n$ training data points are equally distributed to the $D$ blocks for training. On another front, too small $K$ may result in insufficient Adam steps, while too large $K$ may over-optimize one block before moving to others. Therefore, one possible choice is

$$K = \min\left\{\max\left\{\frac{n}{BD}, 50\right\}, 100\right\}. \tag{7}$$

  The suggested selection is merely a guideline. One may also choose other reasonable values for $K$ based on $\frac{n}{BD}$, once each of the $D$ blocks is appropriately trained with a reasonable amount of data.

- For applying LoRA to finetune the Llama 2-7B model, we set LoRA rank to be 100 so that it has as many trainable parameters as BAdam at each iteration. However, when finetuning the Llama 3-8B model, using rank 100 results in out-of-memory error, as shown in Table 2. We observe that LoRA with rank 8 and rank 100 achieve similar MT-bench scores for finetuning the Llama 2-7B model, corroborating the conclusion in LoRA paper [22, Section 7.2] that the LoRA rank does not essentially affect its performance. Therefore, we report the performance of LoRA with rank 8 for finetuning the Llama 3-8B model.

| Hyperparameter | Value |
|---|---|
| lr | 1e-5 |
| lr scheduler | linear decay (lr_min = 0) |
| warm up ratio | 0.1 |
| bz | 16 |
| epoch | 32 |
| weight decay | 0.01 |
| $K$ in BAdam | 100 |
| LoRA rank | 8 |
| LoRA alpha | 4×LoRA rank |

Table 8: Hyperparameters of SuperGLUE tasks.

| Model | Hyperparameter | Value |
|---|---|---|
| Llama 2-7B | lr | 1e-5 and 1e-6 |
| | lr scheduler | cosine (lr_min = 0) |
| | bz (LOMO) | 8 |
| | bz (Other methods) | grad. accu. (2) × single pass bz (8) = 16 |
| | epoch | 3 |
| | weight decay | 0.01 |
| | $K$ in BAdam | 100 |
| | LoRA rank | 100 |
| | LoRA alpha | 4×LoRA rank |
| | Galore rank | 256 |
| | Galore subspace change freq. | 256 |
| | Galore scale factor | 0.25 |
| Llama 3-8B | lr | 1e-5 and 1e-6 |
| | lr scheduler | cosine (lr_min = 0) |
| | bz (LOMO) | 4 |
| | bz (Other methods) | grad. accu. (8) × single pass bz (2) = 16 |
| | epoch | 3 |
| | weight decay | 0.01 |
| | $K$ in BAdam | 100 |
| | LoRA rank | 8 |
| | LoRA alpha | 4×LoRA rank |
| | Galore rank | 256 |
| | Galore subspace change freq. | 256 |
| | Galore scale factor | 0.25 |

Table 9: Hyperparameters of instruction tuning.

| Model | Hyperparameter | Value |
|---|---|---|
| Llama 3-8B | lr (BAdam, Galore, LoRA) | 1e-5 |
| | lr (LOMO) | 1e-4 |
| | lr (Adam) | 1e-6 |
| | lr scheduler | cosine (lr_min = 0) |
| | bz (LOMO) | 8 |
| | bz (Other methods) | grad. accu. (2) × single pass bz (8) = 16 |
| | epoch | 3 |
| | weight decay | 0.01 |
| | $K$ in BAdam | 100 |
| | LoRA rank | 8 |
| | LoRA alpha | 4×LoRA rank |
| | Galore rank | 256 |
| | Galore subspace change freq. | 256 |
| | Galore scale factor | 0.25 |
| Llama 3-70B | lr (BAdam, LoRA) | 1e-5 |
| | lr scheduler | cosine (lr_min = 0) |
| | bz | grad. accu. (8) × single pass bz (2) = 16 |
| | epoch | 3 |
| | weight decay | 0.01 |
| | $K$ in BAdam | 100 |
| | LoRA rank | 8 |
| | LoRA alpha | 4×LoRA rank |
| | Galore rank | 256 |
| | Galore subspace change freq. | 256 |
| | Galore scale factor | 0.25 |

Table 10: Hyperparameters of math finetuning.

# C   Additional Experiment Results

## C.1   Ablation Study on Ordering Strategy and Switching Frequency

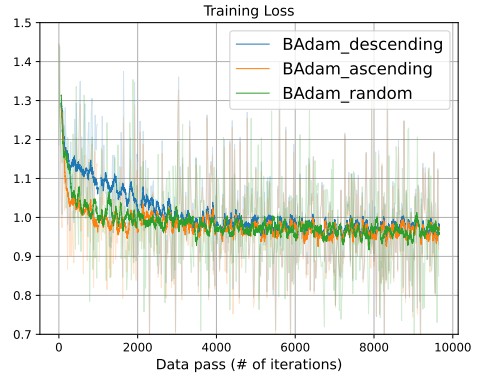 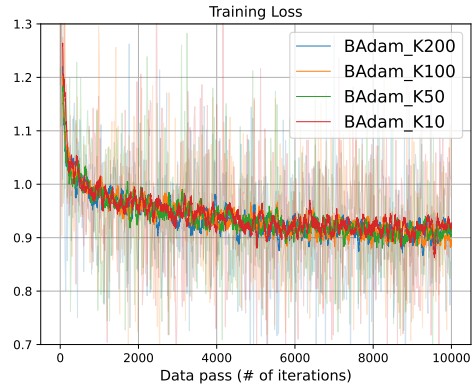

(a) Loss of different ordering strategies.  (b) Loss with different Adam steps $K$.

Figure 4: Effect of ordering strategies and Adam steps $K$.

We display the online loss under different block switch strategies and inner Adam step choices for each block sub-problem in Figure 4. The results are based on finetuning Llama 3-8B on Alpaca-GPT4 dataset.

We compare three block ordering strategies including descending (from the output to the input module), ascending (from the input to the output module), and random reshuffling. As shown in Figure 4a, although the descending scheme initially converges more slowly, all three ordering schemes finally exhibit nearly identical convergence behaviors. As shown in Figure 4b, different choices of Adam steps $K$ does not affect the convergence of online training loss evidently for the task of finetuning Llama 3-8B. However, we notice that the convergence speed may vary across different models for different values of $K$. One can refer to (7) for a detailed discussion on selecting $K$.

## C.2   Experiments on Natural Language Understanding

| Method | BoolQ | COPA | WSC | RTE | MultiRC | WiC |
|--------|-------|------|-----|-----|---------|-----|
| Adam | **0.86** | 0.59 | **0.68** | **0.87** | 0.76 | **0.70** |
| LoRA | 0.81 | 0.56 | 0.62 | 0.79 | 0.69 | 0.59 |
| BAdam | 0.85 | **0.69** | 0.65 | 0.76 | **0.77** | 0.64 |

Table 11: SuperGLUE benchmark scores of the finetuned RoBERTa-large using different optimization methods.

We test the performance of BAdam on classification tasks by training RoBERTa-large [32] on the SuperGLUE benchmark [56]. The implementation is based on jiant [47] using a single RTX3090. The setting of hyperparameters is put in Appendix B.2. We display the test results on 6 tasks selected from the SuperGLUE benchmark. We choose these tasks to conduct experiments since they are selected in [37, 38]. The results can be found in Table 11. It can be observed that BAdam outperforms LoRA in 5 out of the 6 tasks. Furthermore, BAdam demonstrates performance that is comparable to, or tied with, Adam. Based on these results, we can conclude that BAdam is capable of closing the performance gap with Adam more efficiently than LoRA. Consequently, we extrapolate that BAdam has the potential to perform nearly as well as Adam, even when finetuning larger models.

### C.3 Continue Pretrain Llama 3.1-8B-Instruct on StarCoder Dataset

We conduct a preliminary continue pretraining experiment on StarCoder-Python [28] dataset, which consists of repositories from GitHub, including GitHub issues and commits. We train Llama 3.1-8B-Instruct on Starcoder dataset using BAdam, with learning rate 1e-5 and batch size 160. The training loss is shown in Figure 5. One can see that BAdam effectively decreases the CPT loss to around 0.89 after being trained for 10 billion tokens (about 1 epoch).

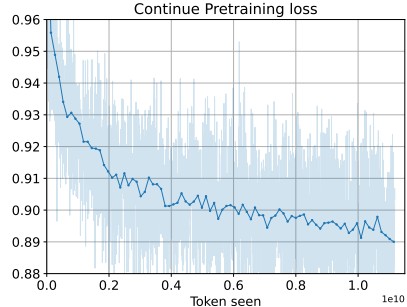

Figure 5: Loss of continue pretrain Llama 3.1-8B-Instruct on StarCoder-Python dataset using BAdam.

### C.4 Memory Consumption and Running Time for Finetuning Llama 2-7B

We present the memory consumption for finetuning the Llama 2-7B model in Table 12. We can see that all the methods successfully finetune Llama 2-7B within 24GB memory, and the cost for each part corroborates our interpretation in Section 2.2.

We also display the time costs for finetuning the Llama 2-7B model in Table 13 and Table 14. One can see that LoRA requires approximately twice the forward time cost when fine-tuning the Llama 2-7B model. This additional cost may be attributed to implementation-level overhead. Notably, finetuning the Llama 2-7B model requires a much longer training time compared to that of the Llama 3-8B model, as the latter utilizes grouped query attention that significantly improves both inference and backward efficiency.

| Method | Parameter | Gradient | Optim. states | Memory consump. |
|---|---|---|---|---|
| Adam | 13.4GB | 26.8GB | 80.4GB | 120.6GB+ |
| LOMO | 13.4GB | 0.4GB | — | 18.8GB |
| LoRA-rank100 | 14.0GB | 1.0GB | 3.0GB | 22.1GB |
| LoRA-rank8 | 13.5GB | 0.1GB | 0.2GB | 20.1GB |
| BAdam | 13.4GB | 0.8GB | 2.4GB | 21.8GB |

Table 12: Actual memory costs of applying mixed precision training to finetune Llama 2-7B with gradient checkpointing using a single RTX3090. Note that LOMO only supports FP16 precision training. The maximum input sequence length is 728 and the batch size is 8.

| Method | Forward | Backward | Update |
|---|---|---|---|
| LOMO | 1.35 hours | 9.71 hours | — |
| LoRA | 2.48 hours | 9.45 hours | 56 seconds |
| BAdam | 1.16 hours | 5.54 hours | 39 seconds |

Table 13: Time spent per epoch on forward, backward, and update for finetuning Llama 2-7B using a single RTX3090. The single pass batch size is 8. The results are averaged over 3 epochs.

| Backward scheme | Backward time |
|---|---|
| All modules | 5.180 seconds |
| Input module only | 3.903 seconds |
| Output module only | 0.053 seconds |

Table 14: Time spent on different backward schemes with batch size 8 for finetuning Llama 2-7B using a single RTX3090. The results are averaged over 100 backward passes.

# D  Convergence Analysis

To establish the convergence of BAdam, we first prove a descent inequality for updates applied to one block by bounding the error terms introduced by Adam updates. Integrating these descent inequalities across different blocks shows that BAdam is a descent method and has a complexity bound of $\mathcal{O}(\delta^{-2})$. For a compact and clear convergence analysis, we focus on BAdam with deterministic gradients, leaving the stochastic cases for future work.

We make the following two assumptions. Assumption D.1 is standard for analyzing block coordinate descent-type methods [58]. Assumption D.2 is commonly used in the analysis of Adam [13]. We adopt this assumption for simplicity of presentation, noting that it can be provably ensured [27].

**Assumption D.1.** The loss function $\mathcal{L}$ is $L$-smooth. And when restricted on $i$-th block, it is $L_i$-smooth. Mathematically,

$$\|\nabla\mathcal{L}(\theta^1) - \nabla\mathcal{L}(\theta^2)\| \leq L\|\theta^1 - \theta^2\|, \quad \left\| \frac{\partial\mathcal{L}}{\partial\theta_i}\bigg|_{\theta_i^1} - \frac{\partial\mathcal{L}}{\partial\theta_i}\bigg|_{\theta_i^2} \right\| \leq L_i\|\theta_i^1 - \theta_i^2\|, i = 1, \ldots, D.$$

We define parameters $\bar{L} = \max_{i=1,\ldots,D} L_i$ as the maximum smoothness constants across all blocks and $\underline{L} = \min_{i=1,\ldots,D} L_i$.

**Assumption D.2.** BAdam has bounded partial derivatives along its trajectory, i.e., there exists $G > 1$ such that

$$\|g_i^{t,k}\| \leq G.$$

Here, we adopt the notations as specified in Algorithm 1: $t = 0, \ldots, T$ are block epochs, $i = 1, \ldots, D$ are different blocks, $k = 0, \ldots, K$ are inner iterations over data for a certain block, and $g_i^{t,k} = \nabla_i\mathcal{L}(\theta_i^t) = \frac{\partial\mathcal{L}}{\partial\theta_i^{t,k}}$ is the partial derivative w.r.t. the $i$-th block at $t$-th block epoch and $k$-th inner Adam step. To avoid potential confusion, in this section we denote $\lambda$ as the numerical stability constant used in the denominator in Adam's adaptive step sizes (Line 12 of Algorithm 1) instead of $\varepsilon$, as the $\varepsilon$ is a conventional notation used to represent the target accuracy in optimization community.

**Corollary D.3** (bounded adaptive step sizes). *Let* $H_i^{t,k} = diag\left(1/\left(\sqrt{\hat{v}_i^{t,k}} + \lambda\right)\right)$ *denote the diagonal matrix formed by coordinate-wise adaptive step sizes vector. Under Assumption D.2 and with $0 < \lambda < 1$, we have:*

$$\frac{1}{2G}I \preccurlyeq H_i^{t,k} \preccurlyeq \frac{1}{\lambda}I.$$

*Proof.* By definition of $\hat{v}$, its elements $(\hat{v}_i^{t,k})_j$ are exponential moving average of the history gradients' squared elements $(g_i^{t,k})_j^2$, so we have bound

$$\max_{1 \leq j \leq d}(\hat{v}_i^{t,k})_j \leq \max_{1 \leq j \leq d}(g_i^{t,k})_j^2 \leq \|g_i^{t,k}\|^2 \leq G^2.$$

Therefore

$$\frac{1}{\sqrt{(\hat{v}_i^{t,k})_j} + \lambda} \geq \frac{1}{G + \lambda} \geq \frac{1}{2G}, \quad \forall 1 \leq j \leq d,$$

then $H_i^{t,k} - \frac{1}{2G}I$ is a positive semidefinite matrix. Similarly, since $1/(\sqrt{(\hat{v}_i^{t,k})_j} + \lambda) \leq 1/\lambda$, we have $H_i^{t,k} \preccurlyeq \frac{1}{\lambda}I$. □

Here we formally present the theorem for the convergence of BAdam in Section 2.1. This section will focus on providing the proof for the theorem.

**Theorem D.4** (descent method). *Under Assumption D.1 and Assumption D.2 and suppose that $0 < \lambda < 1$. If the learning rate satisfies the inequality $\alpha \leq \frac{\lambda}{2\bar{L}K}\min\left\{\frac{1}{K}, \frac{\lambda}{12G}\right\}$, then BAdam with deterministic gradients has the following descent property after one block-epoch of updates:*

$$\mathcal{L}(\theta^{t+1}) - \mathcal{L}(\theta^t) \leq -\frac{\alpha K}{16G\left(1 + \frac{2\alpha^2 KL^2 D}{\lambda^2}\left(\frac{4\bar{L}^2\alpha^2 K^3}{\lambda^6} + 1\right)\right)}\|\nabla\mathcal{L}(\theta^t)\|^2.$$

**Corollary D.5** (first-order complexity). *Under the conditions in Theorem D.4 and setting the learning rate as $\alpha = \frac{\lambda}{4\bar{L}KD^{1/4}} \min\left\{\frac{1}{KD^{1/4}}, \frac{\lambda}{6G}\right\}$, BAdam find a $\delta$-approximate stationary point with at most*

$$T = \frac{128D^{1/4}\bar{L}G(\mathcal{L}(\theta_0) - \mathcal{L}^*)}{\delta^2\lambda} \max\left\{D^{1/4}K, \frac{6G}{\lambda}\right\}$$

*gradient evaluations.*

*Proof.* With the above choice of learning rate and Theorem D.4, the descending property of one block epoch can be written as

$$\mathcal{L}(\theta^{t+1}) - \mathcal{L}(\theta^t) \leq -\frac{\alpha K}{32G}\|\nabla\mathcal{L}(\theta^t)\|^2.$$

Sum over $t$, we obtain the bound for minimum gradient norm

$$\min_{t \leq T}\|\nabla\mathcal{L}(\theta^t)\|^2 \leq \frac{1}{T+1}\sum_{t=0}^{T}\|\nabla\mathcal{L}(\theta^t)\|^2 \leq \frac{32G(\mathcal{L}(\theta_0) - \mathcal{L}^*)}{\alpha K}.$$

Substitute the choice of $\alpha$ and we get the complexity result. $\qquad\square$

To simplify notation, in the following we abuse notation $\mathcal{L}(\theta_i^t)$ to represent $\mathcal{L}\left(\theta_1^{t+1}, \ldots, \theta_{i-1}^{t+1}, \theta_i^{t+1}, \theta_{i+1}^t, \ldots, \theta_D^t\right)$. And $\bar{\theta}_i^t$ represents $\left(\theta_1^{t+1}, \ldots, \theta_{i-1}^{t+1}, \theta_i^{t+1}, \theta_{i+1}^t, \ldots, \theta_D^t\right)$.

**Lemma D.6** (approximate descent inequality for one block). *Under the conditions in Theorem D.4, we have the following approximate descent property for the inner solver of Adam:*

$$\mathcal{L}(\theta_i^t) - \mathcal{L}(\theta_{i-1}^t) \leq -\frac{\alpha K}{2G}\left(\frac{1}{2} - \frac{2L_i\alpha KG}{\lambda^2}\right)\|\nabla_i\mathcal{L}(\theta_i^t)\|^2 + \left(\alpha G + L_i\alpha^2 K^2\right)\|e_i^t\|^2,$$

*where we denote $\|e_i^t\| = \|\frac{1}{K}\sum_{k=1}^{K}\frac{1}{1-\beta_1^k}H_i^{t,k}(m_i^{t,k} - (1-\beta_1^k)g_i^{t,1})\|$ as the difference between the updates of Adam and full GD scaled by coordinate wise adaptive step sizes.*

*Proof.* With Assumption D.1, we have the following descent inequality:

$$\mathcal{L}(\theta_i^t) - \mathcal{L}(\theta_{i-1}^t) \leq \langle\nabla_i\mathcal{L}(\theta_i^t), \theta_i^{t+1} - \theta_i^t\rangle + \frac{L_i}{2}\|\theta_i^{t+1} - \theta_i^t\|^2$$

$$\leq -\alpha\langle\nabla_i\mathcal{L}(\theta_i^t), Ke_i^t + \sum_{k=0}^{K}H_i^{t,k}g_i^{t,1}\rangle + L_i\alpha^2\left(K^2\|e_i^t\|^2 + \left\|\sum_{k=1}^{K}H_i^{t,k}g_i^{t,1}\right\|^2\right)$$

$$\leq -\frac{\alpha K}{2G}\|\nabla_i\mathcal{L}(\theta_i^t)\|^2 + \frac{\alpha K}{4G}\|\nabla_i\mathcal{L}(\theta_i^t)\|^2 + \alpha KG\|e_i^t\|^2$$

$$\quad + L_i\alpha^2 K^2(\frac{1}{\lambda^2}\|\nabla_i\mathcal{L}(\theta_i^t)\|^2 + \|e_i^t\|^2)$$

$$= -\frac{\alpha K}{2G}\left(\frac{1}{2} - \frac{2L_i\alpha KG}{\lambda^2}\right)\|\nabla_i\mathcal{L}(\theta_i^t)\|^2 + \left(\alpha KG + L_i\alpha^2 K^2\right)\|e_i^t\|^2,$$

where the last inequality is because Corollary D.3 and Cauchy-Schwarz inequality. $\qquad\square$

We further have the following lemma, which provides a bound for the above error term.

**Lemma D.7** (bound for error term). *Under the conditions in Theorem D.4, we have bound for $\|e_i^t\|$:*

$$\|e_i^t\| \leq \frac{2L_i\alpha K}{\lambda^2}\|g_i^{t,1}\|.$$

*Proof.* By definition in Algorithm 1 we have the expression for inner updates:

$$m_i^{t,k} = \beta_1 m_i^{t,k-1} + (1-\beta_1)g_i^{t,k} = (1-\beta_1)\sum_{j=1}^{k}\beta_1^{k-j}g_i^{t,j}. \tag{8}$$

So the error term can be written as:

$$\|e_i^t\| = \left\| \frac{1}{K} \sum_{k=1}^{K} \frac{1}{1-\beta_1^k} H_i^{t,k}(m_i^{t,k} - (1-\beta_1^k)g_i^{t,1}) \right\|$$

$$\leq \frac{1}{K} \sum_{k=1}^{K} \frac{1}{\lambda(1-\beta_1^k)} \|m_i^{t,k} - (1-\beta_1^k)g_i^{t,1}\|$$

$$\leq \frac{1}{K} \sum_{k=1}^{K} \frac{1}{\lambda} \|(1-\beta_1) \sum_{j=1}^{k} \beta_1^{k-j}(g_i^{t,j} - g_i^{t,1})\|$$

$$\leq \frac{1-\beta_1}{K\lambda} \sum_{j=2}^{K} \|g_i^{t,j} - g_i^{t,1}\| \sum_{k=j}^{K} \beta_1^{k-j}$$

$$\leq \frac{L_i}{K\lambda} \sum_{j=1}^{K-1} \|\theta_i^{t,j} - \theta_i^{t,0}\|,$$

where the first inequality is because Corollary D.3 and the second inequality is by definition of $m_i^{t,k}$ in (8).

Denote $\Delta_i^t = \sum_{j=1}^{K-1} \|\theta_i^{t,j} - \theta_i^{t,0}\|$, now let's bound $\Delta_i^t$.

$$\Delta_i^t = \sum_{j=1}^{K-1} \|\theta_i^{t,j} - \theta_i^{t,0}\|$$

$$= \sum_{j=1}^{K-1} \alpha \left\| \sum_{k=1}^{j} H_i^{t,k} \hat{m}_i^{t,k} \right\|$$

$$\leq \sum_{j=1}^{K-1} \alpha \sum_{k=1}^{j} \frac{1}{\lambda(1-\beta_1^k)} \left\| (1-\beta_1) \sum_{l=1}^{k} \beta_1^{k-l} g_i^{t,l} \right\|$$

$$= \sum_{j=1}^{K-1} \alpha \sum_{k=1}^{j} \frac{1}{\lambda(1-\beta_1^k)} \left\| (1-\beta_1) \sum_{l=1}^{k} \beta_1^{k-l}(g_i^{t,l} - g_i^{t,1}) + (1-\beta_1^k)g_i^{t,1} \right\|$$

$$\leq \sum_{j=1}^{K-1} \sum_{k=1}^{j} \frac{\alpha}{\lambda(1-\beta_1^k)} \left( (1-\beta_1) \sum_{l=1}^{k} \beta_1^{k-l} L_i \|\theta_i^{t,l} - \theta_i^{t,0}\| + (1-\beta_1^k)\|g_i^{t,1}\| \right)$$

$$\leq \sum_{j=1}^{K-1} \sum_{k=1}^{j} \frac{\alpha}{\lambda(1-\beta_1^k)} \left( (1-\beta_1) L_i \Delta_i^t \sum_{l=1}^{k} \beta_1^{k-l} + (1-\beta_1^k)\|g_i^{t,1}\| \right)$$

$$\leq \frac{\alpha}{\lambda} L_i K^2 \Delta_i^t + \frac{\alpha}{\lambda} K^2 \|g_i^{t,1}\|.$$

In the above, the first inequality is because Corollary D.3 and the second inequality is by Assumption D.1. Therefore $\frac{\alpha K^2}{\lambda}\|g_i^{t,1}\| \geq (1 - \frac{\alpha L_i K^2}{\lambda})\Delta_i^t$ and since $\alpha \leq \frac{\lambda}{2L_i K^2}$ we get $\Delta_i^t \leq 2\frac{\alpha}{\lambda}K^2\|g_i^{t,1}\|$. Further we have

$$\|e_i^t\| \leq \frac{2L_i \alpha K}{\lambda^2} \|g_i^{t,1}\|.$$

$\square$

**Corollary D.8** (refined descent inequality for one block). *The approximate descent inequality for one block in Lemma D.6 can be refined as*

$$\mathcal{L}(\theta_i^t) - \mathcal{L}(\theta_{i-1}^t) \leq -\frac{\alpha K}{8G} \|\nabla_i \mathcal{L}(\theta_i^t)\|^2 \tag{9}$$

*Proof.* Substitute Lemma D.7 into Lemma D.6 and we have

$$\mathcal{L}(\theta_i^t) - \mathcal{L}(\theta_{i-1}^t) \leq -\frac{\alpha K}{2G}\left(\frac{1}{2} - \frac{2L_i \alpha K G}{\lambda^2} - \frac{8L_i^2 \alpha^2 K^2 G^2}{\lambda^4} - \frac{8L_i^3 \alpha^3 K^3 G}{\lambda^4}\right)\|\nabla_i\mathcal{L}(\theta_i^t)\|^2$$

$$\leq -\frac{\alpha K}{8G}\|\nabla_i\mathcal{L}(\theta_i^t)\|^2,$$

where the last inequality is because $\alpha \leq \frac{\lambda^2}{24\bar{L}KG}$. $\qquad\square$

Now we are ready to prove Theorem D.4.

*Proof of Theorem D.4.* Sum up (9) over $i$, we have

$$\mathcal{L}(\theta^{t+1}) - \mathcal{L}(\theta^t) \leq -\frac{\alpha K}{8G}\sum_{i=1}^{D}\|\nabla_i\mathcal{L}(\theta_i^t)\|^2. \tag{10}$$

In the following, we use the right hand side $\sum_{i=1}^{D}\|\nabla_i\mathcal{L}(\theta_i^t)\|^2$ to upper bound the whole gradient vector of one block epoch $\|\nabla\mathcal{L}(\theta^t)\|^2$.

$$\|\nabla_i\mathcal{L}(\theta^t)\|^2 \leq 2\|\nabla_i\mathcal{L}(\theta_i^t)\|^2 + 2\|\nabla_i\mathcal{L}(\theta^t) - \nabla_i\mathcal{L}(\theta_i^t)\|^2$$

$$\leq 2\|\nabla_i\mathcal{L}(\theta_i^t)\|^2 + 2\|\nabla\mathcal{L}(\theta^t) - \nabla\mathcal{L}(\bar{\theta}_i^t)\|^2 \tag{11}$$

$$\leq 2\|\nabla_i\mathcal{L}(\theta_i^t)\|^2 + 2L^2\|\bar{\theta}_i^t - \theta^t\|^2,$$

where the second inequality is because the partial derivative vector $\nabla_i\mathcal{L}(\theta_i^t)$ is part of the gradient vector $\nabla\mathcal{L}(\bar{\theta}_i^t)$ and the last inequality is by $L$-smoothness of $\mathcal{L}$.

For the second term we have

$$\|\bar{\theta}_i^t - \theta^t\|^2 = \sum_{j=1}^{i}\|\theta_j^{t+1} - \theta_j^t\|^2$$

$$= \sum_{j=1}^{i}\alpha^2\left\|Ke_j^t + \sum_{k=0}^{K}H_j^{t,k}g_j^{t,1}\right\|^2$$

$$\leq \sum_{j=1}^{i}2\alpha^2 K^2\|e_j^t\|^2 + \sum_{j=1}^{i}\frac{2\alpha^2 K^2}{\lambda^2}\|\nabla_j\mathcal{L}(\theta_j^t)\|^2$$

$$\leq \sum_{j=1}^{i}\frac{2\alpha^2 K}{\lambda^2}\left(\frac{4L_i^2\alpha^2 K^3}{\lambda^6} + 1\right)\|\nabla_j\mathcal{L}(\theta_j^t)\|^2.$$

Substitute into (11) and sum over, we have

$$\|\nabla\mathcal{L}(\theta^t)\|^2 = \sum_{i=1}^{D}\|\nabla_i\mathcal{L}(\theta^t)\| \leq 2\left(1 + \frac{2\alpha^2 KL^2 D}{\lambda^2}\left(\frac{4\bar{L}^2\alpha^2 K^3}{\lambda^6} + 1\right)\right)\sum_{i=1}^{D}\|\nabla_i\mathcal{L}(\theta_i^t)\|^2.$$

Substitute back into (10) and we get our descent inequality for a whole block epoch. $\qquad\square$

