# OpenReview forum: "BAdam: A Memory Efficient Full Parameter Optimization Method for Large Language Models"
_NeurIPS.cc/2024/Conference — NeurIPS 2024 poster_

### Official Review · Reviewer_JX1Q · 2024-06-19

**Soundness:** 3
**Presentation:** 3
**Contribution:** 3
**Rating:** 8
**Confidence:** 4

**Summary:**

This paper proposes BAdam, a novel optimization method for memory-efficient full parameter finetuning of large language models. BAdam leverages the block coordinate descent framework with Adam as the inner solver. It partitions the model parameters into blocks and updates one block at a time using Adam steps.

**Strengths:**

1. The paper addresses the highly relevant and important problem of enabling full parameter finetuning of large language models under memory constraints. BAdam provides an original and creative solution by combining block coordinate descent with Adam.
2. The theoretical convergence analysis lends credibility to the proposed method, even if limited to the deterministic case. The proof seems sound and complete.
3. The experiments are thorough and convincing, clearly demonstrating BAdam's advantages in memory efficiency, running time, convergence behavior, and downstream performance compared to strong baselines like LoRA and LOMO across multiple models and datasets.

**Weaknesses:**

1. While sufficient for an initial proposal, the theoretical analysis is limited to the deterministic case. Extending the convergence results to the stochastic setting would further strengthen the paper.
2. The paper focuses on applying BAdam to the finetuning stage of large language models. It would be interesting to explore and discuss whether the proposed method could also be applied during the pretraining phase. While I understand that conducting experiments on pretraining may be prohibitive due to time constraints, providing some conceptual discussion on the feasibility, potential benefits, and challenges of using BAdam for pretraining could broaden the scope and impact of the work. For example, the authors could comment on whether the block-wise update scheme of BAdam would remain effective and efficient when dealing with the larger datasets and longer training horizons typically involved in pretraining. Addressing this aspect, even briefly, would give readers a more comprehensive view of BAdam's potential across different stages of the model development pipeline.

**Questions:**

See my weakness part.

**Limitations:**

The authors have discussed key limitations such as the theoretical analysis being restricted to deterministic gradients and comparisons with Adam being limited to medium-sized models due to resource constraints. Suggestions are provided to address these limitations in future work.

---

> ### Author Rebuttal · Authors · 2024-08-07
>
> Thank you very much for the constructive and helpful comments. We address the reviewer's concerns in a point-by-point manner below. *All additional experiment results (figures and tables) are put in the one page supplementary PDF of the global rebuttal.*
>
> **A. Convergence result under stochastic setting.** Our approach of obtaining complexity under the deterministic setting consists of two main steps: (1) establishing the descent property for the inner solver within one block and (2) aggregating the descent inequalities across all blocks for one block-epoch. Step (2) is a standard technique in the complexity analysis of BCD-type methods under reasonable conditions. To extend to stochastic setting, the main difficulty lies in extending step (1). The complete proof is beyond the scope of this rebuttal stage; we provide a proof sketch here. We adopt the same assumptions on the smoothness of the objective function and stochastic gradient errors as in [1].
>
> *Notation modification.* We distinguish the true gradient $g_{i}^{t,k}$ and the estimated stochastic gradient $\tilde{g}\_i^{t,k}$. The difference term $e_{i}^{t}$ between the true gradient $g_{i}^{t,k}$ and the scaled momentum $\hat{m}_{i}^{t,k}$ now includes stochastic gradient errors:
>
> $$
> e_{i}^{t, k} = \hat{m}_{i}^{t, k} - g\_{i}^{t, k},
> $$
>
> where the momentum is  updated with stochastic gradients $m\_{i}^{t,k} = \beta\_{1}  m\_{i} ^{t,k-1}  + (1- \beta\_{1} ) \tilde{g}\_{i} ^{t,k} $.
>
> Assuming stochastic gradients with a uniform almost sure bound $\sigma$, similar to [1], $||e_{i} ^{t,k}||$ can be bounded by the norms of the true gradients with probability at least $1- \delta$:
> $$\sum_{k=1}^{K}  ||e_{i} ^{t,k} ||^2 \le \Theta (\lambda) \sum_{k=1}^{K} ||g_i^{t,k}  || ^2 + \mathcal{O}\left(\sigma ^2 \log (\delta ^{-1} ) ( 1/ \beta_1 + \beta_1 K) \right).$$
>
> *Approximate descent inequality (Inner solver).* Substituting the above probabilistic bound of error terms into the following approximate descent inequality for each step of the inner solver:
>
> $$  \mathcal{L}( \theta _{i}^ {t,k}) - \mathcal{L}(\theta _{i} ^{t,k-1} ) \le - \Theta (\alpha ) ||\nabla _{i} \mathcal{L} (\theta _{i} ^{t,k-1} ) || ^2 + \Theta (\alpha) || e\_{i} ^{t,k-1} || ^2.$$
>
> Then combining a telescoping argument with our technique of bounding gradient differences in the inner loop $||g_{i} ^{t,k} - g_{i} ^{t,1} ||, \forall k \le K $ by $||g_{i} ^{t,1} || $:
>
> $$   \sum_{k=1}^{K}  || g _{i} ^{t,k} - g _{i} ^{t,1} || \le \mathcal{O} \left(\alpha K^2 \right) || g _{i} ^{t,1} || + \mathcal{O} \left(\alpha K^{3 /2} \sigma \sqrt{(1 / \beta _{1} + \beta _{1} K) \log (\delta ^{-1} ) } \right). $$
>
> We obtain the following approximate descent inequality for each block under the stochastic setting:
>
> $$\mathcal{L} (\theta\_{i} ^{t} ) - \mathcal{L}(\theta\_{i-1}^{t}  ) \le - \Theta ( \alpha K)  || \nabla_i \mathcal{L}(\theta_i^{t}) ||^2 +\mathcal{O}\left(\alpha ^3 K ^3 \sigma ^2 \log (\delta^{-1} ) ( 1/ \beta_1 + \beta_1 K) \right).$$
>
> *Approximate descent inequality (BCD epoch).* By aggregating the above approximate descent inequality across different blocks, similar to our arguments in the submitted manuscript, we derive the approximate descent inequality for one block-epoch:
>
> $$\mathcal{L}(\theta ^{t+1} ) - \mathcal{L}(\theta ^{t} ) \le - \Theta (\alpha K) ||\nabla \mathcal{L}(\theta ^{t} )|| ^2 + \mathcal{O}\left(\alpha ^3 K ^3 D\sigma ^2 \log (\delta ^{-1} ) ( 1/ \beta_1 + \beta_1 K) \right).$$
>
> Through standard manipulations, we should obtain a complexity result for finding an $\varepsilon$-stationary point.
>
> [1] Li, H., Rakhlin, A., \& Jadbabaie, A. (2023). Convergence of Adam Under Relaxed Assumptions. NeurIPS.
>
> **B. Initial continual pretraining experiment using BAdam.** Pretraining from scratch is out of scope of this work; we have conducted an initial continual pretraining (CPT) experiment for Llama 3-8B-Instruct on the StarCoder dataset using BAdam. We evaluate its performance using the online CPT loss; see Figure (f). We are only able to finish about 0.1 epoch (1.5B tokens) training due to time constraints. The online CPT loss shows that the model is effectively learning new domain specific (code-related) knowledge. In the following, we list several possible reasons why BAdam may become a candidate optimizer for LLM continual pretraining.
>
> *BAdam exhibits as high rank update as that of Adam.* In the pretraining stage, model acquires massive knowledge over different domains by training on billions of tokens. Intuitively, new factual information (not appeared in the pretraining corpus) might not be encoded by low rank update. As shown in Figure (e), BAdam achieves almost the same high rank update as Adam through all modules of different transformer layers, partially ensuring BAdam’s learnability.
>
> *Possible ability of avoiding general knowledge forgetting.* Perhaps one of the most challenging tasks in CPT is to avoid forgetting of general knowledge. We suspect that BAdam might be good at avoding forgetting compared to Adam. This conjecture stems form the fact that the BAdam uses each data batch to update only one block of parameters, thereby better preserving the general knowledge of the model during CPT. This is partly confirmed by that Llama 3-8B-Instruct has a 67.7 MMLU score (0-shot), while the MMLU score of our CPTed checkpoint after training on 1.5B tokens only decreases to 66.7 (without using any model merging techniques).
>
> We hope that our response is satisfactory to the reviewer and that all concerns have been addressed appropriately. We will properly add the above continual pretraining experiments (with additional comparisons with LoRA and Adam) and try to add convergence result under stochastic setting (if our sketch does not contain vital error) to our next manuscript version. If the reviewer has any additional concerns, please let us know during the reviewer-author discussion period.

---

> ### Comment · Reviewer_JX1Q · 2024-08-09
>
> Thanks for the author's detailed response. It addressed most of my concerns. However, I share the same issue as Reviewer XVND regarding the GSM8K and MATH scores. I hope this can be addressed properly.

---

> > ### Author Response · Authors · 2024-08-09
> > **Authors' Response**
> >
> > Thank you for thoroughly reviewing our response and tracking the feedback of other reviewers. We are pleased to know that most of your concerns have been addressed. We have provided further clarification regarding the score gap of GSM8K and MATH, please see our response to reviewer XVND.
> >
> > We would like to thank the reviewer once again for supporting our work. Should you have any additional questions, please let us know.

---

> > > ### Author Response · Authors · 2024-08-13
> > > **Comment on Scores of GSM8K and MATH**
> > >
> > > Dear reviewer JX1Q,
> > >
> > > Thank you for your thoughtful participation in discussion. We would like to provide additional clarifications regarding the few-shot benchmark scores for GSM8K and MATH, as previously elaborated in our response to Reviewer XVND.
> > >
> > > We recognized that these benchmarks are highly sensitive to the chain-of-thought (CoT) prompting technique, where minor variations in the prompts can lead to heavy fluctuations in the score. Consequently, the impact of the optimization algorithms diminishes when CoT samples are employed, as we have empirically verified. Based on this observation, we believe that evaluating the efficacy of BAdam solely based on these specific few-shot scores—without accounting for the CoT's impact—may not provide a convincing assessment of its performance. Additionally, we remark that it would be inappropriate to disregard BAdam's performance on the zero-shot setting, as it provides a measure that excludes the CoT's effects.
> > >
> > > We hope that our clarification addresses your concern regarding the benchmark scores. Should you have any additional question, please let us know.
> > >
> > > Best,
> > > Authors

---

> ### Comment · Reviewer_JX1Q · 2024-08-13
> **Thank you**
>
> Thank you for following up and the detailed response. I appreciate the effort authors put into addressing this issue. Your explanation has increased my confidence in my comment, and I have raised my confidence level from 3 to 4 and score from 6 to 8. My expertise lies more in pretraining language models, and our team also struggles with dealing with the memory issues brought about by the Adam optimizer. Being able to optimize Adam is a good design. From a practical perspective, I believe this paper can bring relatively significant benefits to the field. I hope the authors can open-source their code in a high-quality manner to facilitate language model community use.

---

> ### Author Response · Authors · 2024-08-14
> **Thank You**
>
> The authors would like to express their deepest gratitude to the reviewer for acknowledging our work and responses. We also greatly appreciate the reviewer for increasing the confidence level from 3 to 4 and the score from 6 to 8.
>
> For pretraining, we will complete the continual pretraining experiment to show the efficiency of BAdam in this pretraining-related setting. Regarding open-sourcing, we promise that our implementation code that can reproduce our paper's results will be made publicly available and include the following features:
>
> **1. Easy to use.** The integration of the BAdam and BCD framework into user's own codebase will be straightforward, necessitating minimal changes to the existing code.
>
> **2. Distributed training support.** Our code will support both data-parallel and model-parallel training, based on DeepSpeed ZeRO-3, allowing for the efficient finetuning / training of truly large-scale models (e.g., 70B). We will also make the implementation of distributed training straightforward.
>
> **3. Memory efficiency.** We will ensure that actual memory usage is consistent with the values reported in our paper. For instance, one will be able to train a Llama 3-8B model using a single RTX3090 and a Llama 3-70B model with just $3\times$ A100-80GB GPUs (based on our distributed training implementation).
>
> Once again, we deeply thank the reviewer for your kind words and support of our work.

---

### Official Review · Reviewer_XVND · 2024-07-04

**Soundness:** 3
**Presentation:** 2
**Contribution:** 2
**Rating:** 5
**Confidence:** 4

**Summary:**

The paper introduces BAdam, a memory-efficient optimization method for fine-tuning large language models (LLMs) by leveraging block coordinate descent (BCD) with Adam as the inner solver. BAdam aims to reduce memory consumption while maintaining or improving performance. The paper presents theoretical convergence analysis and experimental results showing BAdam's effectiveness compared to existing methods like LoRA and LOMO, particularly in terms of memory efficiency and downstream performance.

**Strengths:**

- **Extensive Theoretical Proof of Convergence**: The paper provides substantial theoretical evidence to support the convergence of the proposed method.
- **Detailed Method Analysis**: The analysis of the method is thorough, covering aspects like memory consumption and computation time.

**Weaknesses:**

- **Need for More Quantitative Results**: The evaluation of 7B and 8B models requires more quantitative results, such as mathematical and world knowledge benchmarks (e.g., GSM8K and MMLU). Relying solely on MT-bench, which is scored by GPT-4, is not sufficiently objective.
- **Block-wise vs. Layer-wise Updates**: The paper's core discussion revolves around block-wise updates, but the actual implementation uses layer-wise updates. Other formats of block-wise updates should be explored.
- **Similarity to Existing Work**: The motivation of this paper is similar to "LIFT: Efficient Layer-wise Fine-tuning for Large Model Models"[1] which also discusses "learning one layer/block at a time" in Section 3.2. This similarity needs to be addressed and differentiated.
- **Verbose Section 3.1.2**: The discussion in Section 3.1.2 is overly verbose. Experiments with learning rates and other hyperparameters could be moved to the ablation studies or the appendix, while the main text should focus on the core experimental results.

[1] https://openreview.net/forum?id=u0INlprg3U

**Questions:**

N/A

---

> ### Author Rebuttal · Authors · 2024-08-07
>
> Thank you very much for the constructive and helpful comments. We address the reviewer's concerns in a point-by-point manner below. All additional experiment results (figures and tables) are put in the one page supplementary PDF of the global rebuttal.
>
> **A. More quantitative results on MMLU and math benchmarks.** Following the reviewer's suggestion, we have tested the MMLU scores of the instruction-tuned models; see Table 3. We can observe that BAdam performs as good as Adam and outperforms LoRA.
>
> In Table 2, we have also conducted math-instruction tuning for Llama 3.1-8B on the MathInstruct dataset using BAdam and LoRA, and tested several math benchmarks including GSM8K, MATH, NumGLUE, SVAMP, MMLU-Math, and SAT-Math. The results demonstrate that BAdam clearly outperforms LoRA. We have not conducted the same experiments for Adam due to time constraints, but we will add such experiments in the next version.
>
> We believe that these additional quantitative results, together with the MT-bench results, illustrate the capability of BAdam for LLM finetuning.
>
> **B. More block partition schemes.** As requested by the reviewer, we have tested the ratio-based block partition scheme, where one block is formed by selecting a small part of parameters from each matrix module. We can observe that ratio partition performs nearly as well as the layer partition in terms of optimization ability as shown in Figure (c). In addition to this ratio partition, we can also treat each attention / mlp module as one block, and our test shows that this scheme also has similar optimization ability to the layer partition. We will add ablation study on these additional block partition schemes in the next version.
>
> **C. Fundamental differences from LIFT.** Let us first mention that LIFT is properly cited in our Section 4. Though LIFT utilizes a layer-wise update of Adam, it differs from BAdam fundamentally in the following aspects.
>
> *Different principles in algorithm foundation.* LIFT has two types of iteration policies as stated in their Appendix D, including cyclic and grouped iteration policies. The cyclic policy updates a layer for one iteration and then offloads Adam's optimizer states to the CPU (see the last paragraph of their Section 3.2). This is not a valid BCD with Adam scheme because the gradient of the same layer in the next round of update does not match the offloaded Adam's optimizer states (since other layers' parameters have been updated). On the other hand, their grouped policy updates one layer for the total number of scheduled iterations and does not revisit it, and hence there is no BCD loop at all. This makes their grouped policy a special type of block-restricted implementation of Adam rather than a general BCD scheme.
>
> By contrast, BAdam is built on the BCD optimization framework, resulting in a fundamentally different algorithm. BAdam includes an outer BCD loop and properly accumulates the first and second momentum for updating a block ($K$ in our algorithm description). Importantly, when BAdam revisits a block, *its first and second momentum start at 0* (no offloads), which is crucial for ensuring both theoretical and empirical convergence.
>
> *Theoretical convergence guarantee.* Thanks to BAdam's foundation in the BCD framework, we can provide a convergence guarantee, adding reliability and predictability to its performance.
>
> *Practical code implementation.* We have made significant efforts to develop a compatible and robust code implementation for BAdam, with special attention to memory management. For example, our submitted code allows efficient finetuning of Llama 3-8B using a single RTX3090-24GB GPU, whereas LIFT requires several A100 GPUs (according to their experiment description since they have not open-sourced their implementation). Indeed, implementing a direct block-restricted update of Adam could be trivial with sufficient resources, as one can simply set a certain layer to require gradients in the original Adam optimizer without needing complicated memory management and optimizer coding. Our practical implementation ensures broader applicability and benefits for practitioners.
>
> **D. Verbose experiment section.** We fully agree with the reviewer's concern. We have conducted additional experiments, including more quantitative results (MMLU and math benchmarks), ablation experiments comparing Adam, SGD, BAdam, and BSGD (BCD with SGD), comparison with Galore, and continual pretraining; see the one page supplementary PDF of the global rebuttal. We will incorporate these experiments into our next manuscript version. Specifically, our plan is to move a large part of Section 3.1.1, the convergence verification in Section 3.1.2, and the entire Section 3.2 to the Appendix. Instead, we will add more downstream evaluations (such as MMLU and math tests), create an ablation study section, and a continual pretraining section.
>
> We hope that our response is satisfactory to the reviewer and that all concerns have been addressed appropriately. If the reviewer has any additional concerns, please let us know during the reviewer-author discussion period.

---

> > ### Comment · Reviewer_XVND · 2024-08-08
> >
> > Thank you for the authors' response.
> > We have reviewed your feedback and noted that most of our concerns have been addressed. However, we observed an issue with your Table 2 and Table 3 in the supplementary PDF. According to the official reports for LLaMA 3 and LLaMA 3.1 [1,2], the baseline performance that you report is lower than the official figures, especially for GSM8K and MATH in Table 2. The official tech report [1] indicates that LLaMA 3.1 8B achieves **84.5** on GSM8K (8-shot, CoT), whereas your report shows **17.8**. Similarly, the official MATH (0-shot, CoT) score is **51.9**, while your report indicates **8.6**. For Table 3, the tech report [2] shows that LLaMA 3 8B scores **66.7** on MMLU (5-shot). Based on this information, I believe your supplementary PDF lacks credibility and may negatively impact your paper.
> >
> >
> >
> >
> > > Reference:
> > > [1] https://ai.meta.com/blog/meta-llama-3-1/
> > > [2] https://huggingface.co/meta-llama/Meta-Llama-3.1-8B-Instruct

---

> > > ### Author Response · Authors · 2024-08-08
> > > **Authors' Reponse**
> > >
> > > Thank you for reading our rebuttal and confirming that most of your concerns have been addressed. We now provide clarifications in response to your further questions regarding the gap in evaluation scores.
> > >
> > > **A. Lower baseline GSM8K and MATH (in Table 2) than Meta's official values.**  In the sequel, [1] and [2] refer to the two references the reviewer provided. The values reported in [1] are based on the finetuned model, i.e., Llama 3.1-8B-Instruct. This can be confirmed by comparing several benchmark scores in [1] with those of the instruct version in [2]. Namely, Meta just uses Llama 3.1-8B to represent Llama 3.1-8B-Instruct in [1]. In our provided Table 2 in the supplementary PDF, the "Base model" refers to the *pretrained base model*, i.e., Llama 3.1-8B. In addition, all of the reported scores in our Table 2 are obtained using *0-shot* prompting, which can also be different from that of 8-shot. Hence, we have different baseline performance from Meta's official values. In our Table 2, we use the evaluation code open-sourced by the MathInstruct paper [3] and use the same evaluation setup for all the models (base and finetuned models), ensuring a fair comparison between different optimization methods.
> > >
> > > **B. Lower baseline MMLU (in Table 3) than Meta's official value.** The few-shot MMLU score heavily depends on the *prompt engineering* and *evaluation approach*, as revealed by the Hugging Face report [4]. In this report, it explains in detail why open LLM leaderboard has a much lower MMLU score of the Llama model compared to the official one released by Meta. Since Meta does not open-source their prompts and evaluation methodology that yields their reported MMLU score, the open-source evaluation code may produce a lower MMLU score than Meta's official value. In our provided Table 3 in the supplementary PDF, we use the open-sourced MMLU evaluation code from Llama-Factory [5] and use the same evaluation setup for all the models (base and finetuned models).
> > >
> > > We would like to thank the reviewer once again for reading our rebuttal and raising further questions. We hope that the above response clarifies our results. If the reviewer has any additional questions, please let us know.
> > >
> > > **References:**
> > >
> > > [3] Yue et al. "Mammoth: Building math generalist models through hybrid instruction tuning", ICLR 2024.
> > >
> > > [4] Fourrier et al. "What's going on with the Open LLM Leaderboard?", Hugging Face Blog.
> > >
> > > [5] Zheng et al. "Llamafactory: Unified efficient fine-tuning of 100+ language models", ACL 2024.

---

> > > > ### Comment · Reviewer_XVND · 2024-08-08
> > > >
> > > > Thank you for your detailed response.
> > > >
> > > > A: According to the Llama3 paper [6], Table 12, the Llama3-8B (the pre-trained model) achieves a score of **57.2** on GSM8K and 20.3 on MATH. Consequently, the Llama3.1-8B should exhibit performance that is at least equal to or greater than these values, which are still higher than the scores you have reported.
> > > >
> > > > B: I appreciate your clarification on this point and am satisfied with your response regarding the MMLU scores.
> > > >
> > > > Thank you once again for your thorough explanations.
> > > >
> > > >
> > > > > References:
> > > > > [6] The Llama 3 Herd of Models, https://arxiv.org/pdf/2407.21783

---

> > > > > ### Author Response · Authors · 2024-08-09
> > > > > **Authors' Response**
> > > > >
> > > > > Thank you for your careful reading of our response. We are pleased to see that your concern regarding the MMLU score has been addressed. We now provide further clarifications regarding the gap in the baseline GSM8K and MATH scores.
> > > > >
> > > > > Let us first remark that all the values reported in Llama 3 paper are for Llama 3.1 models, as indicated in [6, paragraph 3 of page 1]. Hence, the reported values in its Table 12 are for Llama 3.1-8B instead of Llama 3-8B. Importantly, their 57.2 GSM8K score is achieved using the *8-shot chain-of-thought prompting* technique, and their 20.3 MATH score is obtained through the *crafted 4-shot math prompting* technique. These setups can be found  in [6, paragraph "Experiment setup" of page 29], in which they refer to their GitHub evaluation page. In summary, they use tailored techniques for almost each task to obtain the officially reported values. Instead, we use the default *0-shot standard prompting* setting for all the benchmarks, i.e., no additional prompts are added to the original input.  The use of carefully designed chain-of-thought prompting may significantly enhance the benchmark scores. For instance, as displayed in the seminal chain-of-thought paper [7, Figure 2], the GSM8K score increases from 0.18 to 0.57 after the chain-of-thought prompting technique is applied to the same model.
> > > > >
> > > > > We would like to thank the reviewer once again for your thorough review of our results. We hope the response clarifies the gap in evaluation scores. If the reviewer has any additional questions, please let us know.
> > > > >
> > > > > **References:**
> > > > >
> > > > > [7] J Wei, et al. "Chain-of-Thought Prompting Elicits Reasoning in Large Language Models", NeurIPS 2022.

---

> > > > > > ### Comment · Reviewer_XVND · 2024-08-09
> > > > > >
> > > > > > Thank you for your thorough response.
> > > > > >
> > > > > > If possible, could you please conduct a new inference experiment that aligns with the official benchmark settings? I believe this process would not take much time and would provide results that are more fair and convincing. When I first raised this concern, it should have prompted a reassessment of the experiment's validity and whether it was fair and meaningful, rather than attempting to convince the reviewer of an evidently questionable result.

---

> > > > > > > ### Author Response · Authors · 2024-08-10
> > > > > > > **Authors' Response**
> > > > > > >
> > > > > > > | Shot       | 0     | 1     | 2     | 3     | 4     | 5     | 6     | 7     | 8     | Average        |
> > > > > > > |------------|-------|-------|-------|-------|-------|-------|-------|-------|-------|----------------|
> > > > > > > | Base model | 17.8  | 46.9  | 45.0  | 53.0  | 54.2  | 51.2  | 55.6  | 54.9  | 56.7  | 48.4           |
> > > > > > > | LoRA       | 48.7  | 61.0  | 57.2  | 59.0  | 59.7  | 58.7  | 60.1  | 59.4  | 59.8  | 58.2           |
> > > > > > > | BAdam      | 49.6 | 60.5  | 61.9  | 59.6  | 59.4  | 59.1  | 61.6  | 61.0  | 59.2  | 59.1       |
> > > > > > >
> > > > > > > **Table 1: GSM8K score.**
> > > > > > >
> > > > > > > ---
> > > > > > >
> > > > > > > | Shot  | 0     | 1     | 2     | 3     | 4     | Average |
> > > > > > > |-------|-------|-------|-------|-------|-------|---------|
> > > > > > > | Base  | 8.6   | 15.7  | 19.7  | 19.7  | 19.8  | 16.7    |
> > > > > > > | LoRA  | 13.7  | 22.8  | 23.2  | 23.0  | 22.8  | 21.1    |
> > > > > > > | BAdam | 17.2 | 23.2  | 22.8  | 22.5  | 22.8  | 21.7 |
> > > > > > >
> > > > > > > **Table 2: MATH score.**
> > > > > > >
> > > > > > > ---
> > > > > > >
> > > > > > > | Shot        | 0     | 1     | 2     | 3     | 4     | 5     | 6     | 7     | 8     | Average        |
> > > > > > > |-------------|-------|-------|-------|-------|-------|-------|-------|-------|-------|----------------|
> > > > > > > | Base model  | 17.8  | 46.9  | 54.8  | 53.9  | 51.7  | 55.6  | 54.2  | 55.6  | 53.4  | 49.3           |
> > > > > > > | LoRA        | 48.7  | 61.0  | 64.9  | 63.6  | 62.5  | 61.9  | 62.2  | 62.5  | 60.7  | 60.9           |
> > > > > > > | BAdam       | 49.6 | 60.5  | 64.1  | 61.5  | 61.0  | 62.9  | 64.4  | 62.7  | 62.5  | 61.0       |
> > > > > > >
> > > > > > > **Table 3: GSM8K score after swapping the second and eighth CoT examples.**
> > > > > > >
> > > > > > > We apologize for misunderstanding the reviewer's request and greatly appreciate the suggestion regarding the experiment setup.
> > > > > > >
> > > > > > > We use the same prompt examples and template from Meta's GitHub instructions to conduct evaluations on the GSM8K and MATH benchmarks. The results are presented in Table 1 and Table 2. The scores for *all shots* are displayed for a comprehensive study.
> > > > > > > The base model achieves a 56.7 GSM8K score with 8-shot CoT, and a 19.8 MATH score with 4-shot CoT, which nearly match Meta's official values of 57.2 and 20.3, respectively.
> > > > > > >
> > > > > > > Although BAdam slightly outperforms LoRA on average, the scores of all three models vary across the $n$-shot results ($n \geq 1$), particularly for GSM8K. This indicates that more shots might not necessarily lead to better results, and adding one more CoT example (i.e., one more shot) may even negatively influence the outcome. Such variability leads us to question the stability of the metric and motivates us to examine the influence of different prompting strategies. To that end, we conduct a small ablation study on the GSM8K benchmark. Specifically, we alter the ordering of the prompts by swapping the second CoT example with the eighth, without changing any other content of these CoT examples. Surprisingly, *this seemingly minor change results in substantial score fluctuations*, as shown in Table 3. Note that this change is not due to randomness, as we have fixed the random seed and use greedy decoding rather than sampling during inference. Therefore, two tests should generate exactly the same scores once the evaluation setup is unchanged. Compared to Table 1, the overall performance of all three models clearly increases, as verified by the average scores. Additionally, LoRA performs almost as well as BAdam in this case.
> > > > > > >
> > > > > > > **Conclusions.** Based on these additional results, we make the following *preliminary* conclusions:
> > > > > > >
> > > > > > > * Compared to the base model, proper finetuning can improve the scores of benchmarks like GSM8K and MATH.  However, under the few-shot CoT setting, we suspect that the dominant factor determining the score is the CoT examples and even the ordering of these examples, rather than the different finetuning optimization methods. That is, the effects of different optimization methods (such as LoRA and BAdam) diminish when few-shot CoT is utilized. This can be partly indicated by comparing the performance of model tuned by LoRA or BAdam with that of the base model under the 0-shot and few-shot settings.
> > > > > > >
> > > > > > > * Since the 0-shot setting excludes the effect of CoT, it could provide a simpler setup for comparing different optimization methods. In this case, BAdam appears to perform better than LoRA.
> > > > > > >
> > > > > > > We deeply thank the reviewer for the insightful discussion and constructive feedback, which helped us better understand these benchmarks. We hope these additional results address the reviewer's concerns. If the reviewer has any additional questions and/or suggestions, please let us know.

---

> > > > > > > > ### Comment · Reviewer_XVND · 2024-08-10
> > > > > > > >
> > > > > > > > Thank you for your detailed response.
> > > > > > > >
> > > > > > > > Given the limited performance improvement, I have decided to maintain my current score.

---

### Official Review · Reviewer_9PUh · 2024-07-12

**Soundness:** 2
**Presentation:** 3
**Contribution:** 3
**Rating:** 5
**Confidence:** 3

**Summary:**

The paper introduces memory-efficient optimizer BAdam, which combines the concepts of block coordinate descent (BCD) and Adam's update rule. BAdam demonstrates that, with moderate memory consumption—more than LOMO—it can surpass LoRA and significantly outperform LOMO in fine-tuning Llama 2-7B and Llama 3-8B. Additionally, BAdam shows similar fine-tuning performance to Adam when applied to the medium-sized masked language model RoBERTa-large.

**Strengths:**

The paper's contributions and strengths are as follows:

1. It proposes using the well-known optimization technique BCD for the task of fine-tuning large language models while being memory efficient.
2. Empirical evidence highlights BAdam's potential in fine-tuning language models. It can outperform LoRA and significantly exceed LOMO in instruction-tuning Llama 2-7B and Llama 3-8B. Moreover, BAdam demonstrates comparable fine-tuning performance to Adam when used with the medium-sized masked language model RoBERTa-large.
3. It provides theoretical convergence analysis for the deterministic case.

**Weaknesses:**

The paper's weaknesses are summarised as follows:

1. In extremely memory-limited settings, such as when there is only enough memory for inference, where MeZO or LOMO can apply, BAdam cannot apply due to the additional requirement of storing block-wise optimizer states.
2. The paper does not provide theoretical or practical insights into why BCD is effective for language model fine-tuning.
3. Compared to LoRA, BAdam requires storing full parameter checkpoints instead of a small number of adapters. As the scale of language models continues to grow, the issue of storing full parameters for fine-tuning becomes much more intolerable in practice.

**Questions:**

1. In Table 1, why LOMO update precision is only limited to Float16. Can't it be Float32?
2. In Table 5, using the same learning rate for SGD and AdamW is not convincing. Typically, SGD requires a larger learning rate than AdamW to achieve good performance. Have the authors tried conducting a grid search to determine the optimal learning rate for LOMO?
3. When presenting the results in Table 5, what parameters were fixed? Batch size, memory, etc.? I noticed that the batch sizes are not consistent across methods; for example, LOMO uses a batch size of 8, while LoRA and BAdam use a batch size of 16. Comparing different optimization methods with varying batch sizes is not ideal.
4. I believe more ablation studies could be conducted. For example, why is Adam used for the intermediate steps instead of SGD? How much performance loss would occur if SGD were used instead?

**Limitations:**

The author has addressed the limitations.

---

> ### Author Rebuttal · Authors · 2024-08-07
>
> Thank you very much for the constructive and helpful comments. We address the reviewer's concerns in a point-by-point manner below. *All additional experiment results (figures and tables) are put in the one page supplementary PDF of the global rebuttal.*
>
> **A. Ablation study & effectiveness of BCD for LLM finetuning.** We have conducted ablation experiments on Adam, SGD (LOMO), BAdam, and BSGD (BCD with SGD) for finetuning Llama 3-8B.  All optimization methods use grid-searched learning rates based on training and validation losses.
>
> *Convergence ability.* In Figure (b), it can be observed that BCD variants converge slightly slower but soon exhibit similar convergence behavior in terms of running time compared to their full counterparts. It is worth mentioning that, unlike the full counterparts, BCD variants only update a block of parameters per data batch, demonstrating the strong optimization ability of BCD for LLM finetuning.
>
> *Downstream performance.* In Table 1, we also test the MT-bench scores of the finetuned models. It is quite interesting to see that BSGD significantly outperforms SGD (almost as good as BAdam), even though they have almost the same optimization convergence behavior. The superiority of BCD variants over their full counterparts possibly arises from the fact that BCD uses each data batch to update only one block of parameters, thereby better preserving the general knowledge of the pretrained model during finetuning. These strong downstream performances of BCD further illustrate its suitability for LLM finetuning.
>
> *Learnability interpreted by high rankness.* Unlike LoRA and Galore, BCD's memory efficiency is achieved without restricting its updates to a low rank space. In Figure (d), we display the cumulative explained variance of BAdam's update for the up-proj matrix, which is defined as sum of the $k$ largest squared singular value over the sum of all squared singular value. This result shows that BAdam's update has a heavy tailed singular values distribution and is far away from a low rank update. In Figure (e), we show that BAdam achieves almost the same high rank update as Adam through all modules of different transformer layers, partially ensuring BCD's learnability.
>
> We believe that these experiment results demonstrate the effectiveness of BCD for LLM finetuning.
>
> **B. Handle the extremely memory-limited setting with BAdam.** In the extremely memory-limited setting, we have the following two choices to apply BAdam.
>
> *Smaller block partition.* Our BCD optimization framework (as well as our code implementation) provides a flexible block partition. Hence, we can choose a smaller block partition, further reducing the memory consumption as we only need to store the gradient and optimizer states for a smaller block of parameters. We have tested such a smaller block setting of BAdam (each layer is further partitioned to attention and mlp blocks), and it achieves a MT-bench score of 6.50, which is just slightly worse than the score of 6.67 reported in the manuscript.
>
> *Cheap CPU update.* We can also store the optimizer states in CPU memory, and performs the cheap block update purely on CPU. In this case, only the block gradient needs to be communicated between CPU and GPU during update. We have tested this approach under the Llama 3-8B experiment setting and found that the operation only induces a 0.8-second overhead per update, resulting in roughly 16 minutes of additional time cost over 3 epochs of training. We will upload this part of new code implementation when uploading is allowed.
>
> We remark that MeZO and LOMO also maintain the gradient of a certain block in memory when updating this block. Thus, our BCD scheme can be applied using one of the above schemes whenever MeZO and LOMO are applicable.
>
> **C. BAdam needs to store the full checkpoint.** Storing the full parameter checkpoint appears to be inevitable when using full parameter optimization methods such as Adam and SGD. While LoRA has the advantage of storing different type of checkpoints, full parameter finetuning might achieve higher performance, as we have shown.
>
> **D. LOMO update precision issue.** LOMO's update precision depends on the precision of the model weights. Consequently, updating the model using Float32 will double the memory consumption to $4M$. We follow the reported update precision (Float16) used in the LOMO paper.
>
> **E. LOMO's performance under grid searched step size.** We have conducted a grid search for SGD (LOMO)'s learning rate among (5e-2, 1e-2, 1e-3, 1e-4, 1e-5, 1e-6) based on loss, and identified 1e-2 as the optimal one for MT-bench evaluation. In Table 1, it can be observed that SGD (LOMO) with learning rate 1e-2 achieves an MT-bench score of 5.82, which aligns with the score of 5.83 obtained by using learning rate 1e-6 in our manuscript.
>
> **F. Batch size is not consistent across different methods.** Since LOMO performs update on the fly and does not store the full gradient, it does not support gradient accumulation. We choose the largest batch size under the memory constraint of RTX3090-24GB for LOMO. We also ensure each algorithm uses the same amount of data. Hence, we count twice updates of LOMO as one update in our manuscript, ensuring a fair comparison to LoRA and BAdam.
>
> We hope that our response is satisfactory to the reviewer and that all concerns have been addressed appropriately. We will properly add the above experiments to our next manuscript version. If the reviewer has any additional concerns, please let us know during the reviewer-author discussion period.

---

> > ### Comment · Reviewer_9PUh · 2024-08-09
> >
> > The reviewer thanks the authors for the discussion. I decide to keep my score.

---

> > > ### Author Response · Authors · 2024-08-09
> > > **Authors' Response**
> > >
> > > Thank you for your careful review of our paper and your positive assessment. If you have any further questions, please let us know.

---

### Official Review · Reviewer_FWuu · 2024-07-14

**Soundness:** 2
**Presentation:** 2
**Contribution:** 2
**Rating:** 3
**Confidence:** 4

**Summary:**

The authors proposed fine-tuning of LLMs with a block coordinate descent based Adam optimizer.
They presented results on convergence analysis, memory and run time profiling, and the quality of resulting fine-tuned models.

**Strengths:**

+ The proposed idea is clearly stated.
+ The memory usage analysis is comprehensive.
+ The authors conducted experiments on recent LLMs.

**Weaknesses:**

- The novelty of the proposed technique is BCD optimization for fine-tuning LLMs, using the Adam optimizer.  In this sense, the comparison against LOMO is justified, but the empirical advantage is not very significant.  The author did not perform ablation study showing the necessity of the two ingredients: BCD and Adam--another important condition to compare with LOMO and BAdam would be BCD with SGD.
- The comparison against PEFT such as LoRA is not scientifically justified, on the other hand.  Because BCD and parameter-efficient reparameterization are not mutually exclusive, and could be complementary.
- In addition to optimizer (e.g. LOMO and this work) and PEFT, there is a third class of memory-efficient LLM fine-tuning techniques: gradient compression (e.g. Galore, arXiv:2403.03507).  Like LoRA, this is orthogonal and potentially complementary to BCD optimization as well.  There is no comparison here.

**Questions:**

* See above weaknesses on empirical results.
* A crucial motivation of the proposed method, is a conjectured unique suitability of BCD in LLM fine-tuning: as the authors put it, "the finetuning process boils down to an optimization problem that needs to handle a huge number of trainable model parameters, while the number of training data points are relatively small. This setting matches exactly the advantage of the BCD scheme".  Unfortunately, however, this remained a conjecture.  There is a lack of adequate theoretic or empirical support to this interesting and important open question.

**Limitations:**

Yes.

---

> ### Author Rebuttal · Authors · 2024-08-07
>
> Thank you very much for the constructive and helpful comments. We address the reviewer's concerns in a point-by-point manner below. *All additional experiment results (figures and tables) are put in the one page supplementary PDF of the global rebuttal.*
>
> **A. Ablation study on BCD, Adam, and SGD.** As requested by the reviewer, we have conducted experiments on Adam, SGD (LOMO), BAdam, and BSGD (BCD with SGD as inner solver) for finetuning Llama 3-8B. All optimization methods use grid-searched learning rates based on training and validation losses.
>
> *Convergence ability.* In Figure (b), it can be observed that BCD variants converge slightly slower but soon exhibit similar convergence behavior in terms of running time compared to their full counterparts. It is worth mentioning that, unlike the full counterparts, BCD variants only update one block of parameters per data batch, demonstrating the strong optimization ability of BCD for LLM finetuning.
>
> *Downstream performance.* In Table 1, we also test the MT-bench scores of the finetuned models. It is quite interesting to see that BSGD significantly outperforms SGD (almost as good as BAdam), even though they have almost the same optimization convergence behavior. The superiority of BCD variants over their full counterparts possibly stems from the fact that BCD uses each data batch to update only one block of parameters, thereby better preserving the general knowledge of the pretrained model during finetuning. These strong downstream performances of BCD further illustrate its suitability for LLM finetuning.
>
> These ablation results partly confirm the suitability of BCD for LLM finetuning and reveal that choosing either Adam or SGD as the inner solver has similar performance in terms of our downstream performance evaluation (i.e., MT-bench score for model trained on Alpaca-GPT4 dataset). We will add BSGD to our BCD framework and leave the comprehensive study of different inner solvers as future work.
>
> **B. The suitability of BCD in LLM finetuning.** Let us elaborate on the suitability of BCD for LLM finetuning from the following three perspectives.
>
> *Performance.* Based on the results of the above ablation study, we believe that we have demonstrated the strong performance achieved by BCD compared to its full counterpart. This directly indicates that BCD is suitable for LLM finetuning in terms of both optimization ability and downstream performance.
>
> *Low memory consumption.* BCD not only achieves strong performance but is also memory efficient; it only stores the gradient and optimizer states of the active block, which is substantially lower than Adam's $18M$ cost. In particular, BSGD requires little additional memory beyond just storing the $2M$ LLM, while still exhibiting strong downstream performance. Given that memory is usually the bottleneck for finetuning LLM, the memory efficiency of BCD further demonstrates its suitability for this task.
>
> *Learnability interpreted by high rankness.* Unlike LoRA and Galore, BCD's memory efficiency is achieved without restricting its updates to a low rank space. In Figure (d), we display the cumulative explained variance of BAdam's update for the up-proj matrix, which is defined as sum of the $k$ largest squared singular values over the sum of all squared singular values. This result shows that BAdam's update has a heavy tailed singular values distribution and is far away from a low rank update. In Figure (e), we show that BAdam achieves almost the same high rank update as Adam through all modules of different transformer layers, partially ensuring BCD's learnability.
>
> **C. BCD + LoRA.** We first note that the high rank update of BAdam shown in Figures (d) and (e) partly illustrates the scientific difference between BCD and LoRA. Following the reviewer's suggestion, we also combined BCD with LoRA (B-LoRA) for LLM finetuning. Due to the advantages of BCD, B-LoRA can have a higher rank configuration under the same amount of memory budget and save computation time, compared to LoRA.  B-LoRA has a slightly worse MT-bench score than LoRA (6.12 vs. 6.28) as shown in Table 1, while achieves a better 5-shot MMLU score than LoRA as shown in Table 3.
>
> **D. Comparison with Galore.** In Figures (a)-(b) and Table 1, we report the results of Galore with rank 1024 and 8-bit Adam optimizer for finetuning Llama 3-8B. We set other hyperparameters according to their paper's suggestions. It can be seen that BAdam and BSGD outperform Galore in terms of MT-bench downstream performance, demonstrating the ability and suitability of BCD for LLM finetuning. In terms of convergence, BAdam also exhibits better convergence ability than Galore.
>
> We have not conducted experiments on combining BCD and Galore due to time constraints, but we will add such experiments in our next manuscript version, as suggested by the reviewer.
>
> **E. Significance of advantage over LOMO.** We note that the MT-bench score difference between GPT-3.5-turbo and GPT-4 on the LMSYS leaderboard is 1.05, which is close to the gap of 0.96 presented in our manuscript between the Llama 3-8B models tuned by LOMO and BAdam (5.69 versus 6.65). Therefore, the advantage appears to be significant.
>
> We hope that our response is satisfactory to the reviewer and that all concerns have been addressed appropriately. We will properly add the above experiments to our next manuscript version. If the reviewer has any additional concerns, please let us know during the reviewer-author discussion period.

---

> ### Author Response · Authors · 2024-08-13
> **Looking forward to your feedback (if any)**
>
> Dear reviewer FWuu,
>
> We hope that our response justifies the design of BAdam and addresses most of your concerns. Since the deadline of the discussion phase is approaching, we would highly appreciate to receive feedback from you. If you have any further questions or remarks, please let us know so that we will have enough time to address your concerns. We will be more than happy to provide additional clarifications and details.
>
> Best,
>
> Authors.

---

### Author Rebuttal · Authors · 2024-08-07

Dear ACs and Reviewers,

This global response contains our one-page supplementary PDF of the rebuttal. All additional figures  and tables are included in this file.

Best regards,

Authors.

---

### Decision · Program_Chairs · 2024-09-25

**Decision:**

Accept (poster)

**Comment:**

This paper looks into the problem of how to perform memory-efficient fine-tuning of LLMs. LLM training is memory-intensive, due to the large volume of parameters and the memory for storing optimizer states. To address this issue, the paper proposes BADAM, a method that leverages the block coordinate descent framework with Adam as the inner solver. Basically, it partitions the model parameters into blocks and updates one block at a time using Adam steps. By doing this, the max memory footprint of fine-tuning LLMs is reduced.

Overall, this is a borderline paper. While BAdam can potentially become a good alternative way for fine-tuning LLMs with limited memory capacity. There are a few concerns:
1) There have already been many memory-efficient fine-tuning methods developed, such as LoRA, GaLore, etc. Empirical comparison with these methods shows that BAdam does not have significant advantage over those methods despite it supports full-parameter optimization.
2) Full-parameter optimization requires saving checkpoints of the full model states, which is appearing only if the performance improvement is great. However, the improvement seems to be marginal.
3) Optimizations are sensitive to hyperparameter choices. The paper uses pre-selected parameters for alternative baselines. Given that there are many hyperparameter options, it is hard to say the performance improvement is mainly from the BAdam approach itself.
4) Even for full parameter optimization, there are many alternative systems optimizations, such as offloading optimizer states to secondary memory or layer-by-layer fine-tuning. There has been a lack of discussion of the proposed method wrt to those optimizations.

Given these, the paper can be accepted only if there are additional space.